# SNN-Driven Multimodal Human Action Recognition via Sparse Spatial-Temporal Data Fusion

## Abstract

Recent multimodal action recognition approaches that combine RGB and skeleton data have achieved strong performance, but their high computational cost and poor energy efficiency hinder deployment on edge devices. To address these limitations, we propose the first spiking neural network (SNN)-based framework for multimodal human action recognition, to the best of our knowledge, offering an energy-efficient and scalable solution that fuses sparse spatiotemporal data of event cameras and skeletons within a unified spiking architecture. The framework leverages the sparse and asynchronous nature of event and skeleton data and the energy-efficient properties of SNNs. It achieves this through a series of tailored components, including modality-specific feature extraction, a sparse semantic extractor, spiking-based cross-modal fusion via Spiking Cross Mamba, and task-relevant feature compression utilizing a Discretized Information Bottleneck (DIB). To support reproducible evaluation, we further introduce a data construction pipeline that generates temporally aligned event-skeleton pairs from existing RGB-skeleton datasets. Extensive experiments demonstrate that our approach achieves state-of-the-art accuracy among SNNs while significantly reducing energy consumption, providing a practical and scalable solution for neuromorphic multimodal action recognition.

## 1 Introduction

Human action recognition (HAR) is a fundamental task in computer vision with applications in intelligent surveillance, human-computer interaction, and medical rehabilitation Yadav et al. (2021). Existing HAR methods rely on diverse modalities, each with unique advantages and limitationsSun et al. (2022). RGB data, processed through Convolutional Neural Networks (CNN)Yao et al. (2019), 3D Convolutional Networks (C3D)Jiang et al. (2020); Zhou et al. (2018); Liu et al. (2018), or Vision Transformers (ViT)Yang et al. (2022), effectively captures spatial-temporal features. However, they are highly sensitive to lighting conditions, background clutter, and occlusions. Shaikh & Chai (2021). Skeleton data, modeled by Graph Convolutional Networks (GCNs)Xin et al. (2023); Yan et al. (2018); Cheng et al. (2020); Zhang et al. (2020) or TransformersAhmad et al. (2021), provides robust geometric information but loses critical details in complex actionsRen et al. (2024). Event cameras, with high dynamic range and low latency, are well-suited for capturing fast movements. Still, they are susceptible to background noise under uncontrolled illumination.de Blegiers et al. (2023); Gallego et al. (2020).

Multimodal human action recognition addresses the limitations of single-modality methods by integrating complementary data sources.Shaikh et al. (2024). Among them, RGB-skeleton fusion, as shown in Figure 1a, has received significant attention, as RGB offers rich visual details while skeleton data enhances structural robustnessAhn et al. (2023); Das et al. (2020); Bruce et al. (2022); Li et al. (2020); Kim et al. (2023). However, they mainly rely on Artificial Neural Networks (ANNs), which operate on floating-point arithmetic, causing high computation and energy costs. RGB dependency also causes storage and memory overhead, especially for high-resolution videos, further raising resource demands. Moreover, multi-stream fusion typically operates at the classification level, often ignoring the intrinsic relationships between fea-

tures. In contrast, event and skeleton data possess sparse characteristics, which reduce memory usage and offer complementary features worth exploring at the feature level for efficient fusion.

Meanwhile, Spiking Neural Networks (SNNs) have emerged as a promising alternative to ANNs. They offer biologically inspired sparse activation and asynchronous processing, which significantly reduce power consumption compared to ANN-based modelsYamazaki et al. (2022); Han & Lee (2021). Recent studies have explored SNNs in single-modality action recognition, including graph-based SNNs for skeleton modelingZheng et al. (2024b) and event-driven SNN architectures for sparse visual processingYao et al. (2024b;a); Zhou et al. (2023); Miki et al. (2023); Ren et al. (2023). Although these approaches demonstrate energy efficiency, they remain limited to single-modality settings, leaving their potential for multimodal fusion largely unexplored. Furthermore, SNNs inherently suffer from information loss due to their discrete spike-based representations, which constrains their ability to encode fine-grained motion details and cross-modal dependencies.

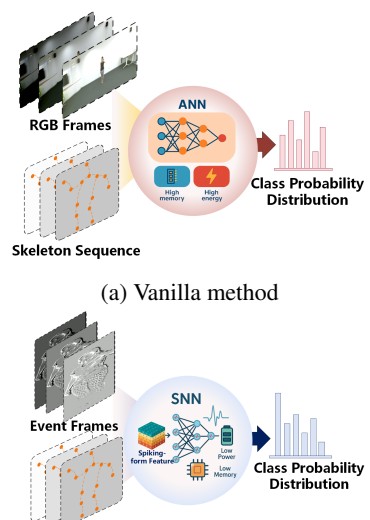

(a) Vanilla method

(b) Our method

Figure 1: Comparison of vanilla ANN-based RGB-Skeleton fusion (a) and our proposed SNN-based event-skeleton fusion (b) for human action recognition.

Building on these insights, we propose the first SNN-driven multimodal framework for human action recognition that jointly models sparse spatiotemporal event vision and skeleton data at the feature level, as illustrated in Figure 1b. This framework is designed to address two core challenges: (i) how to effectively structure and refine sparse spatiotemporal features from each modality within an SNN framework, and (ii) how to retain task-relevant information during fusion despite the quantized, binary nature of spikes. By fusing sparse event and skeleton streams under spiking constraints, our approach better exploits the complementary nature of these modalities, paving the way for more efficient and accurate human action recognition. Our key contributions are summarized as follows:

- **SNN-based multimodal fusion architecture**: We design a unified framework that integrates event and skeleton data within an SNN backbone, incorporating Sparse Semantic Extractor (SSE) and Spiking Cross Mamba (SCM) modules for modality-specific encoding and biologically plausible fusion. This architecture enables efficient and robust action recognition under spiking constraints.

- **Discretized Information Bottleneck (DIB):** We formulate an information-theoretic compression mechanism tailored to the spiking domain, enabling task-relevant feature selection while maintaining compatibility with discrete spike-based computation. DIB significantly enhances the compactness and discriminative power of the fused representations.

- **Data construction and experimental validation:** We develop a reproducible pipeline to synthesize synchronized event-skeleton datasets from RGB-skeleton sources, facilitating comprehensive evaluation. Extensive experiments demonstrate superior accuracy and substantial energy savings, confirming the practicality and scalability of our framework for real-world, resource-constrained applications.

## 2 RELATED WORK

**Multimodal Human Action Recognition** Multimodal human action recognition has emerged as an effective solution to the limitations of single-modality methods by leveraging complementary data sources for improved accuracy and adaptabilitySun et al. (2022); Shaikh et al. (2024). Among fusion strategies, RGB-skeleton integration is widely adopted, combining RGB's rich spatial context with the structured motion cues of skeleton data, enabling robust recognition of both coarse and fine-grained actionsShabaninia et al. (2024); Ren et al. (2021). Numerous studies have demon-

strated that integrating RGB and skeleton data significantly improves recognition robustness, especially in complex environments where single-modality methods struggleAhn et al. (2023); Das et al. (2020); Li et al. (2020); Bruce et al. (2022); Shaikh & Chai (2021); Zhu et al. (2022). Furthermore, recent research has investigated multimodal contrastive learning approaches such as Contrastive Language–Image Pretraining (CLIP), which incorporates textual information to refine vision-based recognition modelsWang et al. (2023a); Qiu & Hou (2024); Wang et al. (2023b). While improving accuracy, existing multimodal approaches still rely heavily on dense data modalities such as RGB, which are not fully optimized for efficient resource usage and energy consumption. In contrast, our method leverages completely sparse spatiotemporal data from both event cameras and skeletons, enabling a more efficient and complementary fusion approach at the feature level.

**Spiking Neural Networks**   SNNs process information via discrete spikes, making them naturally suited for event-driven and energy-efficient computation Dampfhoffer et al. (2023); Yamazaki et al. (2022); Han & Lee (2021). Common neuron models like the Leaky Integrate-and-Fire (LIF) and its variants (e.g., PLIF Fang et al. (2021)) simulate the accumulation of input signals until a threshold triggers a spike, enabling sparse and biologically plausible dynamics Wu et al. (2018). Recent works have applied SNNs to single-modality action recognition — leveraging temporal sparsity in event data de Blegiers et al. (2023); Yao et al. (2024a); Ren et al. (2023); Chen et al. (2024) or structural priors in skeleton-based graphs Zheng et al. (2024b). However, these methods are limited in capturing cross-modal complementarity. To overcome this, we propose a multimodal SNN framework that integrates event and skeleton data, combining structured motion cues with event-driven vision for efficient and robust action recognition.

## 3 METHODOLOGY

An overview of the proposed SNN-Driven Multimodal Human Action Recognition Framework is illustrated in Figure 2. SGN and Spiking Mamba extract features from skeleton and event frames, respectively. The SSE further enhances multimodal representations. SCM enables cross-modal feature interaction, while the DIB preserves essential modality-specific semantics for classification. The following sections detail each module.

### 3.1 SGN AND SPIKING MAMBA: SKELETON AND EVENT ENCODING

As illustrated in the left of Figure 2, our framework adopts SNN to extract structured spiking-form features from both skeleton and event data.

We adapt the spiking graph network (SGN) from prior work on spiking-based graph modeling for skeleton recognition Zheng et al. (2024b) to convert the input skeleton sequence $X_s \in \mathbb{R}^{T \times C \times V}$ into a spiking-form representation. The network applies Linear Projection (LP), Batch Normalization (BN), Spike Neuron (SN), and Spiking Position Embedding (SPE) to transform the features into a spiking-form representation, which is then processed by spiking graph convolution layers to capture spatial dependencies and spiking self-attention (SSA) to model long-range joint interactions. To enhance frequency-domain modeling, we introduce a multi-branch spectral module. The spiking feature is first transformed via FFT, then processed by four parallel branches, each comprising a padded dilated depthwise separable 1D convolution (PDConv1D) with dilation rate $d_i$, followed by BN and SN. The resulting features are then combined and transformed back to the temporal-spatial domain via an Inverse Fast Fourier Transform (IFFT). The SGN can be written as follows:

$$X_s^{(0)} = \text{SN}\left(\text{BN}(\text{LP}(X_s))\right) + \text{SPE}_s, \quad X_s^{(0)} \in \mathbb{R}^{T \times C^{(0)} \times V}, \quad \text{SPE}_s \in \mathbb{R}^{T \times C^{(0)} \times V} \tag{1}$$

$$X_g^{(l)} = \text{SN}\left(\text{BN}\left(\hat{A}^{(l)} X_g^{(l-1)} W_g^{(l)}\right)\right), \quad X_g^{(l)} \in \mathbb{R}^{T \times C^{(l)} \times V}, \quad W_g^{(l)} \in \mathbb{R}^{C^{(l-1)} \times C^{(l)}} \tag{2}$$

$$F_\xi^{(l)} = \text{FFT}(\text{SSA}^{(l)}(X_g^{(l)})), \quad F_\xi^{(l)} \in \mathbb{R}^{T \times C^{(l)} \times V}, \quad \xi \in \{\text{real}, \text{imag}\} \tag{3}$$

$$F_{\xi,d}^{(l)} = \text{SN}(\text{BN}(\text{PDConv1D}(F_\xi^{(l)}, d))), \quad F_{\xi,d}^{(l)} \in \mathbb{R}^{T \times C^{(l)} \times V}, \quad d \in \{1, 2, 3, 4\} \tag{4}$$

$$X_s^{(l)} = \text{IFFT}\left(\|_{d=1}^4 F_{\xi,d}^{(l)}\right), \quad X_s^{(l)} \in \mathbb{R}^{T \times C^{(l)} \times V} \tag{5}$$

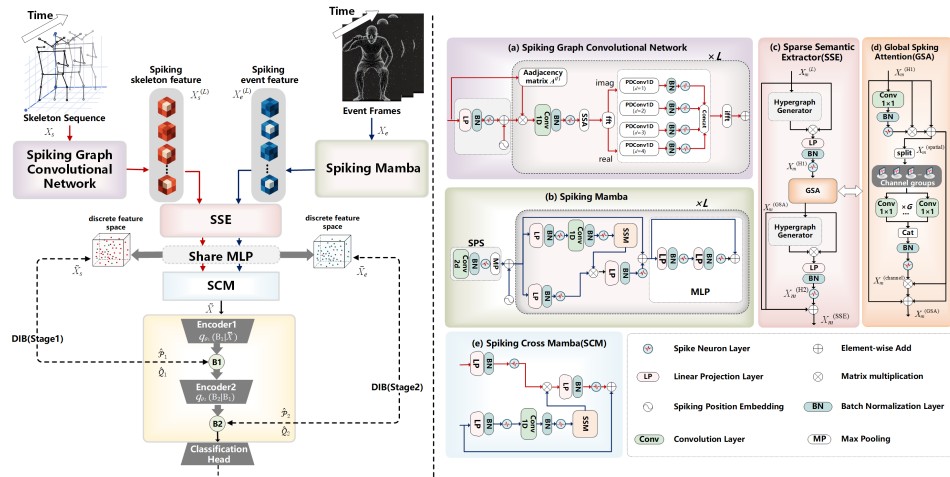

Figure 2: Overview of the proposed SNN-driven multimodal action recognition framework. The left panel shows the full pipeline; right subfigures detail core modules. (a) SGN and (b) Spiking Mamba extract features from skeleton and event data, respectively. (c) SSE enhances modality-specific semantics via structural self-similarity. (d) GSA, within SSE, improves global spiking feature alignment. (e) SCM enables cross-modal interaction through selective and state-space paths. The bottom-left illustrates the two-stage DIB module, which compresses fused features before classification.

After $L$ layers, the final output is $X_s^{(L)} \in \mathbb{R}^{T \times C^{(L)} \times V}$. Here, $T$, $C$, and $V$ denote the temporal length, feature channels, and number of skeleton joints, respectively. $\hat{A}^{(l)} \in \mathbb{R}^{V \times V}$ is the normalized adjacency matrix in the $l$-th graph convolution. $X_g^{(l)}$ and $X_s^{(l)}$ refer to graph-level and spectral-enhanced features, respectively. FFT outputs real and imaginary parts ($\xi \in \{\text{real}, \text{imag}\}$), which are processed separately, and $\|$ denotes channel-wise concatenation.

For the event modality, we first apply a Spiking Patch Splitting (SPS) module Zhou et al. (2023) to obtain spiking patch sequences $X_e^{(0)} \in \mathbb{R}^{T \times V \times D^{(0)}}$, where $V$ denotes the number of patch tokens. The resulting representations are then processed by $L$ stacked Spiking-Mamba blocks Gu & Dao (2023), which adapt the selective state-space model to spiking computation by replacing standard activations with SN. Each block integrates selective and state-space pathways, and incorporates a multilayer perceptron (MLP) built with Linear Projection(LP)-BN-SN layers to refine local features under spiking constraints. This design ensures both long-range dependency modeling and sparse event-driven computation. The final output $X_e^{(L)} \in \mathbb{R}^{T \times V \times D^{(L)}}$ is dimensionally aligned with the skeleton modality for multimodal fusion.

## 3.2 Sparse Semantic Extractor

We propose SSE to enhance the semantic representation of spiking-form skeleton feature $X_s^{(L)}$ and event features $X_e^{(L)}$ independently, as shown in Figure2(c). SSE comprises two key components: Hypergraph Generators (HG) for capturing structured spatiotemporal relationships and a Global Spiking Attention (GSA) module for refining feature importance via attention.

In HG, each spiking-form tensor is first reshaped into a set of $T \cdot V$ spatiotemporal nodes, each with dimensionality $C^{(L)}$. A sparse hypergraph is then constructed by computing the pairwise similarity between nodes. Specifically, for each node $i$, we identify its $k$-nearest neighbors based on Euclidean distance, and define the adjacency matrix $H \in \mathbb{R}^{(T \cdot V) \times (T \cdot V)}$ as:

$$H_{ij} = \frac{1}{1 + \|x_i - x_j\|_2}, \quad \forall j \in \mathcal{N}_k(i), \tag{6}$$

where $x_i$ is the feature vector of the $i$-th node, and $\mathcal{N}_k(i)$ denotes its local neighborhood in the spatiotemporal domain. This adjacency matrix is then used to propagate information among related

nodes via matrix multiplication with the flattened input feature matrix. The result is subsequently passed through LP, BN, and SN to obtain the refined spiking-form feature $X_m^{(\text{H1})} \in \mathbb{R}^{T \times V \times C^{(L)}}$, where $m \in \{s, e\}$.

As illustrated in Figure 2(d), given the input feature $X_m^{(\text{H1})}$, the GSA module applies spatial and channel attention sequentially. In the spatial attention branch, a $1 \times 1$ convolution is first applied to reduce the channel dimension from $C^{(L)}$ to 1. The resulting feature map is then passed through BN and SN, generating an attention map of shape $\mathbb{R}^{T \times V \times 1}$. This attention map is broadcast-multiplied with the input $X_m^{(\text{H1})}$, yielding the spatially attended representation $X_m^{(\text{spatial})}$. Then, in the channel attention branch, $X_m^{(\text{spatial})}$ is divided into $G$ groups along the channel dimension, producing sub-tensors of shape $\mathbb{R}^{T \times V \times \frac{C^{(L)}}{G}}$. Each group is processed independently by a grouped $1 \times 1$ convolution, followed by BN and SN, to capture channel-specific dependencies within each group. The outputs from all groups are concatenated along the channel axis to reconstruct a tensor of shape $\mathbb{R}^{T \times V \times C^{(L)}}$. This aggregated result is matrix multiplied with $X_m^{(\text{spatial})}$ to obtain the channel-attended feature $X_m^{(\text{channel})}$. The spatial and channel-attended features are then fused via residual addition, producing the output $X_m^{(\text{GSA})} \in \mathbb{R}^{T \times V \times C^{(L)}}$. To further capture higher-order structural dependencies, a second hypergraph is constructed, and the same GSA refinement process is applied again, yielding $X_m^{(\text{H2})}$.

Finally, the output of the SSE module integrates multi-stage hypergraph refinement and residual pathways, formulated as:

$$X_m^{(\text{SSE})} = X_m^{(\text{H2})} + X_m^{(\text{GSA})} + X_m^{(L)} \tag{7}$$

This structured attention mechanism enables efficient spiking-based feature extraction and facilitates subsequent multimodal fusion.

## 3.3 MULTIMODAL FUSION MODULES

Before fusing skeleton and event spiking-form features, a shared-weight MLP is applied to align their semantic spaces, producing the transformed representation $\widetilde{X}_m$ in a common feature space while preserving modality-specific characteristics.

### 3.3.1 SPIKING CROSS MAMBA

SCM extends Spiking Mamba by introducing cross-modal interaction between aligned spiking features, as shown in Figure 2(e). In this module, we adapt the two components of the original Mamba architecture: the Selective Path (SP) and the State Space Path (SSP), where SP refines skeleton features ($\widetilde{X}_s$) and SSP captures dependencies in event features ($\widetilde{X}_e$). These paths interact via matrix multiplication to integrate complementary modality features into a unified space. The final multimodal output $\widetilde{X}$, which retains the original event representation through a residual connection to $\widetilde{X}_e$, fuses multimodal information while retaining the sparse, low-power properties of SNNs.

### 3.3.2 DISCRETIZED INFORMATION BOTTLENECK

Information Bottleneck (IB) has shown strong potential in multimodal fusion by enforcing mutual information constraints to retain task-relevant features while discarding redundancy Xiao et al. (2024). However, traditional IB methods rely on continuous activations and reparameterization, which are incompatible with SNNs due to their binary and event-driven nature. This makes mutual information estimation and optimization particularly challenging under spiking constraints. As we further elaborate in Appendix A.1, Gaussian IB formulations fail under spiking sparsity due to ineffective KL regularization, non-differentiable thresholds, unstable Bernoulli reconstruction, and ill-conditioned objectives.

To bridge this gap, we propose the Discretized Information Bottleneck tailored for SNNs. As illustrated in the left-bottom part of Figure 2, DIB adopts a two-stage encoder structure. The first encoder maps $\widetilde{X}$ to $B_1$, applying sparse regularization to maintain modality-awareness. The second encoder compresses $B_1$ into a joint latent representation $B_2$, which is passed to a classification head— the only decoder in the pipeline — for final prediction.

At the first stage, following the information bottleneck principle, we formulate the objective as a constrained optimization problem: $\min_{p(B_1|\widetilde{X}),\, p(B_2|B_1)} I(\widetilde{X}; B_1)$ subject to: $I(\widetilde{X}; \widetilde{X}_s) - I(B_1; \widetilde{X}_s) \leq \epsilon_1, I(B_1; B_2) \leq \epsilon_2, I(\widetilde{X}; \widetilde{X}_e) - I(B_1; \widetilde{X}_e) \leq \epsilon_3$, where $\epsilon_1, \epsilon_2, \epsilon_3 > 0$ control the degree of compression. We reformulate the mutual-information constraints by replacing the compression terms with KL-based regularizers and the retention terms with variational likelihood surrogates. This yields the following tractable objectives:

$$\mathcal{L}_{\mathrm{IB},B_1}^{\theta_1,\psi_1} = I(\widetilde{X}; B_1) - \lambda_1 I(B_1; \widetilde{X}_s)$$

$$\approx \mathbb{E}_{\widetilde{X}\sim P(\widetilde{X})}\, KL\big(q_{\theta_1}(B_1|\widetilde{X}) \,\|\, q(B_1)\big) - \lambda_1 \mathbb{E}_{B_1\sim P(B_1|\widetilde{X})}\mathbb{E}_{\widetilde{X}\sim P(\widetilde{X})}\Big[\log q_{\psi_1}(\widetilde{X}_s|B_1)\Big]. \tag{8}$$

$$\mathcal{L}_{\mathrm{IB},B_2}^{\theta_2,\psi_2} = I(B_1; B_2) - \lambda_2 I(B_2; \widetilde{X}_e)$$

$$\approx \mathbb{E}_{B_1\sim P(B_1)}\, KL\big(q_{\theta_2}(B_2|B_1) \,\|\, q(B_2)\big) - \lambda_2 \mathbb{E}_{B_2\sim P(B_2|B_1)}\mathbb{E}_{B_1\sim P(B_1)}\Big[\log q_{\psi_2}(\widetilde{X}_e|B_2)\Big]. \tag{9}$$

Here $\theta_1$ and $\theta_2$ denote the parameters of encoding neural networks while $\psi_1$ and $\psi_2$ denote the parameters of the output predicting neural networks. $\lambda_1$ and $\lambda_2$ are trade-off parameters. The detailed derivation of this formulation is provided in the Appendix A.2.

To instantiate the KL divergence terms in Eq. 8 and Eq. 9 under spiking constraints, we introduce a discrete Bernoulli KL divergence loss at each compression stage $n \in \{1, 2\}$. Specifically, each encoder consists of two sequential layers. The first is an LP-BN-SN block that encodes the input into a spiking latent representation $B_n$. The second component is an LP-BN layer followed by a sigmoid activation, which transforms the spiking latent representation $B_n$ into Bernoulli-distributed activation probabilities $\hat{\mathcal{P}}_n$, indicating the likelihood of spike activations for each channel. To construct a valid reference prior $\hat{\mathcal{Q}}_n$, we apply an Exponential Moving Average (EMA) over the batch-wise mean of $\hat{\mathcal{P}}_n$, resulting in an input-independent yet sparsity-aware distribution. These two distributions are then used to compute the discrete KL divergence as formulated in Eq. 10:

$$\mathcal{L}_{\mathrm{KL},n} = \hat{\mathcal{P}}_n \log \frac{\hat{\mathcal{P}}_n}{\hat{\mathcal{Q}}_n} + (1 - \hat{\mathcal{P}}_n) \log \frac{1 - \hat{\mathcal{P}}_n}{1 - \hat{\mathcal{Q}}_n}. \tag{10}$$

As standard reparameterization is incompatible with binary spike activations, we adopt a discrete surrogate strategy. Specifically, the latent feature $B_n$ is projected by a learnable matrix $W_n$ and passed through a sigmoid to obtain sampling probabilities. Then, a binary mask $\Gamma_n \sim \mathrm{Bernoulli}(\sigma(W_n B_n))$ is sampled and combined with $B_n$ via bitwise XOR to yield the final representation $\widetilde{B}_n = B_n \oplus \Gamma_n$. This approximates discrete reparameterization within the binary support; gradients through XOR are estimated with a straight-through surrogate. At inference time, the mask is omitted for deterministic decoding.

The retention terms of Eq. 8 and Eq. 9 are expressed as $\lambda_1 \mathbb{E}_{\widetilde{X}\sim P(\widetilde{X})} \mathbb{E}_{B_1\sim P(B_1|\widetilde{X})}\big[\log q_{\psi_1}(\widetilde{X}_s \mid B_1)\big]$ and $\lambda_2 \mathbb{E}_{\widetilde{X}\sim P(\widetilde{X})} \mathbb{E}_{B_1\sim P(B_1|\widetilde{X})} \mathbb{E}_{B_2\sim P(B_2|B_1)}\big[\log q_{\psi_2}(\widetilde{X}_e \mid B_2)\big]$. Since direct variational estimation can be unstable for binary spikes, we approximate these expectations with a *normalized cosine surrogate* (see Appendix. A.3 for assumptions and derivations), leading to the stage-wise objectives:

$$\mathcal{L}_{\mathrm{DIB},1} = \mathcal{L}_{\mathrm{KL},1} - \lambda_1 \widehat{\cos}\big(\psi_1(\widetilde{B}_1),\, \widetilde{X}_s\big), \quad \mathcal{L}_{\mathrm{DIB},2} = \mathcal{L}_{\mathrm{KL},2} - \lambda_2 \widehat{\cos}\big(\psi_2(\widetilde{B}_2),\, \widetilde{X}_e\big), \tag{11}$$

where $\mathcal{L}_{\mathrm{KL},1}$ and $\mathcal{L}_{\mathrm{KL},2}$ are the discrete KL losses for the two compression stages, and $\psi_1(\cdot)/\psi_2(\cdot)$ are spiking MLP projections specialized for skeleton ($\widetilde{X}_s$) and event ($\widetilde{X}_e$), respectively. We define $\widehat{\cos}(a, b) = \left\langle \frac{a}{\|a\|_2+\varepsilon}, \frac{b}{\|b\|_2+\varepsilon} \right\rangle$ with a small $\varepsilon = 1e-6$. This surrogate is heuristic and not a mutual-information bound; see Appendix. A.3 for scope and conditions.

Finally, the fused code $\widetilde{B}_2$ is passed through the classification head to produce $\hat{y}$. The overall training objective is

$$\mathcal{L}_{\mathrm{total}} = \mathcal{L}_{\mathrm{CE}}(\hat{y}, y) + \alpha\big(\mathcal{L}_{\mathrm{DIB},1} + \mathcal{L}_{\mathrm{DIB},2}\big), \tag{12}$$

where $\mathcal{L}_{\mathrm{CE}}(\hat{y}, y)$ is the cross-entropy loss, $\alpha$ controls the trade-off between compression and classification, and $\lambda_1, \lambda_2$ weight the cosine MI surrogates. For completeness, we provide the training pseudocode of DIB in Appendix A.4 and a theoretical feasibility analysis in Appendix A.5.

Table 1: Comparison of Different Models on NTU RGB+D (NRD), NTU RGB+D 120 (NRD120), and NW-UCLA (NU) Datasets. Xs, Xv, and Xt denote the protocols: X-Sub (Cross-Subject), X-View (Cross-View), and X-Set (Cross-Setup), respectively. OPs refers to SOPs in SNN and FLOPs in ANN. E, R, and S denote event data, RGB data, and skeleton sequences, respectively. "–" indicates that the original paper did not provide the corresponding energy formula or accuracy on this dataset, making reproduction or comparison infeasible.

| Model | Modality | Param. (M) | NRD (Xs) | NRD (Xv) | NRD120 (Xs) | NRD120 (Xt) | NU | OPs (G) | Power (mJ) |
|---|---|---|---|---|---|---|---|---|---|
| ST-GCN Yan et al. (2018) | S | 3.1 | 81.5% | 88.3% | 70.7% | 73.2% | - | 3.48 | 16.01 |
| Shift-GCN Cheng et al. (2020) | S | - | 87.8% | 95.1% | 80.9% | 83.2% | 92.5% | 2.5 | 11.5 |
| CTR-GCN Chen et al. (2021) | S | 1.46 | 89.9% | 94.5% | 84.9% | 87.1% | 94.7% | 1.97 | 9.06 |
| Koopman Wang et al. (2023c) | S | - | 90.2% | 95.2% | 85.7% | 87.4% | 95.0% | - | - |
| SGM-NET Li et al. (2020) | S+R | 71.6 | 88.9% | 95.7% | - | - | - | 169.2 | 778.3 |
| MMFF Zhu et al. (2022) | S+R | 29.1 | 89.6% | 96.3% | - | - | - | 24.5 | 112.7 |
| VPN Das et al. (2020) | S+R | 24.0 | 93.5% | 96.2% | 86.3% | 87.8% | - | - | - |
| MMNet Bruce et al. (2022) | S+R | 34.2 | 94.2% | 97.8% | 92.9% | 94.2% | - | 89.2 | 410.32 |
| Spikformer Zhou et al. (2023) | S | 4.78 | 73.9% | 80.1% | 61.7% | 63.7% | 85.4% | 1.69 | 2.17 |
| Spike-driven Transformer Yao et al. (2024b) | S | 4.77 | 73.4% | 80.6% | 62.3% | 64.1% | 83.4% | 1.57 | 1.93 |
| Spike-driven Transformer V2 Yao et al. (2024a) | S | 11.47 | 77.4% | 83.6% | 64.3% | 65.9% | 89.4% | 2.59 | 2.91 |
| Spike Wavelet Transformer Fang et al. (2024) | S | 3.24 | 74.7% | 81.2% | 63.5% | 64.7% | 86.7% | 1.48 | 2.01 |
| STAtten Lee et al. (2025) | S | 3.19 | 72.8% | 79.7% | 60.3% | 61.7% | 82.8% | 1.98 | 2.48 |
| MK-SGN Zheng et al. (2024a) | S | 2.17 | 78.5% | 85.6% | 67.8% | 69.5% | 92.3% | 0.68 | 0.614 |
| Signal-SGN Zheng et al. (2024b) | S | 1.74 | 80.5% | 87.7% | 69.2% | 72.1% | 92.7% | 0.31 | 0.37 |
| Spikformer Zhou et al. (2023) | E | 4.18 | 76.9% | 82.3% | 64.1% | 65.3% | 88.6% | 1.03 | 1.42 |
| Spike-driven Transformer Yao et al. (2024b) | E | 4.48 | 78.3% | 84.5% | 67.4% | 68.3% | 90.3% | 1.00 | 1.37 |
| Spike-driven Transformer V2 Yao et al. (2024a) | E | 6.54 | 82.1% | 89.2% | 70.4% | 71.7% | 93.1% | 9.98 | 11.75 |
| Spike Wavelet Transformer Fang et al. (2024) | E | 3.14 | 78.5% | 84.7% | 69.9% | 70.1% | 91.7% | 1.27 | 1.62 |
| Spikmamba Chen et al. (2024) | E | 3.91 | 78.3% | 83.9% | 69.5% | 70.0% | 90.6% | 0.85 | - |
| STAtten Lee et al. (2025) | E | 4.17 | 79.8% | 84.9% | 67.2% | 69.1% | 91.2% | 0.83 | 1.27 |
| **Ours** | S+E | 7.92 | **85.0%** | **92.3%** | **74.6%** | **76.2%** | **96.7%** | 1.47 | 1.73 |

## 4 EXPERIMENTS

We present evaluations on accuracy and energy consumption based on the constructed event-skeleton datasets. The following sections cover dataset construction, state-of-the-art (SOTA) comparisons, and ablation studies. Detailed experimental settings are provided in the Appendix B, along with additional information on the construction of the data set and visual analysis. We evaluate our framework on three standard benchmarks (NTU RGB+D Shahroudy et al. (2016), NTU RGB+D 120 Liu et al. (2019), and NW-UCLA Wang et al. (2014)). For multimodal training, we further construct event-skeleton pairs on NTU RGB+D by applying ROI-based cropping and converting RGB clips into event streams using V2E Hu et al. (2021); detailed construction steps are provided in Appendix I.

### 4.1 COMPARISONS WITH SOTA

Table 1 presents a comprehensive comparison across three benchmark datasets, evaluating classification accuracy, computational complexity, and energy consumption. To ensure a fair evaluation, we adopt a joint-only input configuration for all skeleton-based models, excluding multi-stream fusion settings (e.g., joint, bone, motion) commonly used in GCN-based approaches. For ANN-based models, we report results from their original papers. For SNN-based models, we apply a unified training pipeline and hyperparameter configuration. Specifically, skeleton-only(Joint) SNN results are either taken directly from Zheng et al. (2024a) or reproduced following the same implementation and settings as described therein. Meanwhile, event-only SNN models are fully reproduced under the same training configuration as our proposed method, ensuring consistent comparison across all spiking models.

Under this unified setup, we measure computational complexity using synaptic operations (SOPs) for SNNs and FLOPs for ANNs, and estimate energy consumption following the methodology described in the Appendix C. Our model achieves remarkable energy efficiency, consuming only 1.73 mJ—an order of magnitude lower than ANN-based models such as MMNet (410.32 mJ) and MMFF (112.7 mJ). Despite leveraging both skeleton and event modalities, our model maintains a compact parameter size of 7.92 M, which is significantly smaller than fusion-based models like SGM-NET (71.6 M) and MMNet (34.2 M), demonstrating strong parameter efficiency. In terms of recognition

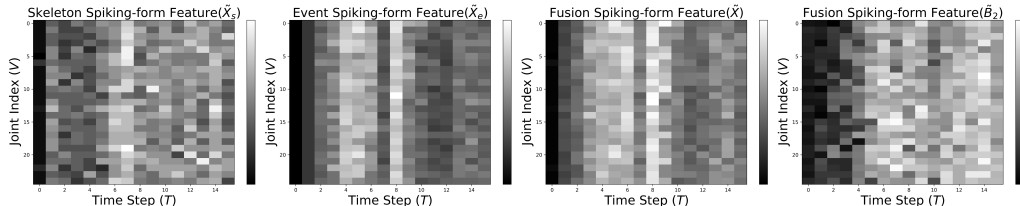

Figure 3: Feature Space Visualization. The grayscale intensity represents the activation level of the spiking neurons at each spatiotemporal point.

performance, our model consistently surpasses all SNN-based methods across the three datasets. On NTU RGB+D 60 (X-view), our approach achieves 92.3% accuracy, significantly higher than both skeleton-based SNNs and event-based SNNs. On NTU RGB+D 120 and NW-UCLA, similar performance gains are observed. Notably, our model attains 96.7% accuracy on NW-UCLA, marking the highest among all compared methods and demonstrating the strength of our spike-driven multimodal fusion framework. For completeness, we further conduct an analysis of multimodal SNN fusion strategies in Appendix G, which corroborates the superior effectiveness of our proposed framework.

## 4.2 ABLATION STUDIES

We conduct a component-wise ablation study on the NTU RGB+D dataset to evaluate each module's contribution in Table 2. Starting from the event-only baseline Spikformer (76.9%), introducing Spiking-Mamba improves accuracy with reduced computation and energy. Adding SGN incorporates the skeleton modality for multimodal learning, significantly boosting performance to 81.3%.

Subsequent modules—SSE, SCM, and DIB—progressively enhance cross-modal representations and promote sparse, task-relevant fusion, culminating in the highest accuracy of 85.0%. These results demonstrate the complementary benefits of each module and the effectiveness of our energy-efficient multimodal fusion. We include class-wise accuracy in the Appendix D for a deeper insight into model behavior.

To better understand our model's spiking dynamics, we visualize joint-time $(V \times T)$ spiking heatmaps of "drinking water" action in Figure 3. The skeleton modality $(\widetilde{X}_s)$ shows structured, stable activations, while the event stream $(\widetilde{X}_e)$ exhibits sharp, scattered temporal spikes, reflecting asynchronous motion sensitivity. The initial fusion output $(\widetilde{X})$ combines complementary traits from both modalities, and the final bottleneck output $(\widetilde{B}_2)$ highlights sharpened, sparser patterns, indicating effective fusion and noise suppression. Additionally, to visualize the discriminative regions learned by the model, we refer to the CAM-Based Skeleton-Level Spatial Interpretation in Appendix H.1, which demonstrates how the model attends to different body regions during action recognition.

Table 2: Component-wise Ablation Study on the NTU RGB+D (Xs) Dataset. $^{*}$ includes the shared-weight MLP layer for feature alignment.

| Method | Param. (M) | Acc (%) | SOPs (G) | Power (mJ) |
|---|---|---|---|---|
| Baseline (Spikformer)[E] | 4.18 | 76.9 | 1.03 | 1.42 |
| Baseline (Signal-SGN)[S] | 1.65 | 78.3 | 0.312 | - |
| Spiking Mamba[E] | 2.58 | 77.1 | 0.94 | 1.13 |
| +SGN[E+S] | 4.71 | 81.3$^{+4.2}$ | 1.16 | 1.25 |
| +SSE[E+S,*] | 6.31 | 82.1$^{+0.9}$ | 1.29 | 1.51 |
| +SCM[E+S] | 6.84 | 82.7$^{+0.6}$ | 1.31 | 1.57 |
| +DIB[E+S] | 7.92 | **85.0**$^{+2.3}$ | 1.47 | 1.73 |

*Note:* The Signal-SGN baseline refers to its backbone part only, for fair comparison.

To further analyze the effectiveness of key components in our framework, we separately evaluate the impact of the HG and the GSA module in SSE at first, with results in Table 3. Both independently improve accuracy and their combination (Full SSE, $k = 3$) further enhances accuracy. The highest accuracy (82.1%) is achieved with $k = 5$, balancing generalization and efficiency, while larger kernels ($k = 7$) show diminishing returns. For further insights into the learned spatiotemporal features, we refer to the Hypergraph Visualization and Semantic Integration in Appendix H.2.

Table 3: Impact of HG and GSA in SSE. * includes the shared-weight MLP layer.

| Method | Param (M) | Acc (%) | SOPs (G) | Power (mJ) |
|---|---|---|---|---|
| Spiking Mamba + SGN | 4.71 | 81.3 | 1.16 | 1.25 |
| +HG (w/o GSA)* | 5.24 | 81.7$^{+0.4}$ | 1.22 | 1.38 |
| + GSA (w/o HG)* | 6.04 | 81.5$^{-0.2}$ | 1.24 | 1.42 |
| + Full SSE($k = 3$)* | 6.31 | 81.7$^{+0.2}$ | 1.27 | 1.45 |
| + SSE ($k = 1$)* | 6.31 | 81.4$^{-0.3}$ | 1.28 | 1.48 |
| + SSE ($k = 5$)* | 6.31 | **82.1**$^{+0.7}$ | 1.29 | 1.51 |
| + SSE ($k = 7$)* | 6.31 | 81.9$^{-0.2}$ | 1.33 | 1.58 |

Table 4: Effect of Modal Exchange in SCM and Stage Selection in DIB.

| Approach | SP | SSP | Stage 1 | Stage 2 | Acc(%) |
|---|---|---|---|---|---|
| Baseline | - | - | - | - | 82.1 |
| +SCM | $\widetilde{X}_e$ | $\widetilde{X}_s$ | - | - | 82.3$^{+0.2}$ |
| +SCM (Final Selection) | $\widetilde{X}_s$ | $\widetilde{X}_e$ | - | - | 82.7$^{+0.4}$ |
| +DIB (Single Stage) | $\widetilde{X}_s$ | $\widetilde{X}_e$ | $\widetilde{X}_s$ | - | 83.5$^{+0.8}$ |
| +DIB (Single Stage) | $\widetilde{X}_s$ | $\widetilde{X}_e$ | $\widetilde{X}_e$ | - | 83.2$^{-0.3}$ |
| +DIB | $\widetilde{X}_s$ | $\widetilde{X}_e$ | $\widetilde{X}_e$ | $\widetilde{X}_s$ | 84.5$^{+1.3}$ |
| +DIB | $\widetilde{X}_s$ | $\widetilde{X}_e$ | $\widetilde{X}_s$ | $\widetilde{X}_e$ | **85.0**$^{+0.5}$ |

Building on this, we next investigate the cross-modal interaction mechanisms, specifically SCM input configuration and DIB stage placement in Table 4. The results evaluate SCM input configuration and DIB stage placement. Using skeleton features as SP input and event features as SSP input yields the best result, confirming the importance of this setup. For DIB, single-stage bottlenecks bring gains, but the two-stage variant with skeleton first and event second achieves the best performance (85.0%), validating sequential refinement.

Having confirmed the optimal structural choices, we then further analyze the robustness of DIB by systematically varying its hyperparameters in Table 5.Removing both IB constraints drops accuracy to 83.2%, showing DIB's impact. Disabling either $\lambda_1$ or $\lambda_2$ leads to smaller gains (83.5%, 83.7%), while balanced tuning with $\alpha = 0.05$, $\lambda_1 = 0.5$, and $\lambda_2 = 0.6$ achieves the best result (85.0%).

These results confirm the importance of joint optimization for effective compression and fusion. Additional evidence, including loss convergence curves in Appendix F and an ablation on alternative DIB formulations in Appendix E, further demonstrates the stability and necessity of our design choices.

| $\alpha$ | 0.05 | 0.05 | 0.05 | 0.03 | 0.04 | 0.05 | 0.05 |
|---|---|---|---|---|---|---|---|
| $\lambda_1$ | 0.0 | 0.0 | 0.5 | 0.5 | 0.5 | 0.5 | 0.5 |
| $\lambda_2$ | 0.0 | 0.6 | 0.0 | 0.6 | 0.6 | 0.6 | 0.4 |
| **Acc(%)** | 83.2 | 83.5 | 83.7 | 83.9 | 84.5 | **85.0** | 84.7 |

Table 5: Hyperparameter ($\alpha, \lambda_1, \lambda_2$) analysis of DIB.

Beyond the quantitative improvements, we provide qualitative evidence through feature visualization. Figure 4 shows the t-SNE distributions across the fusion process. In the first plot, features from individual modalities are poorly separated, indicating limited class separability. As the fusion progresses,

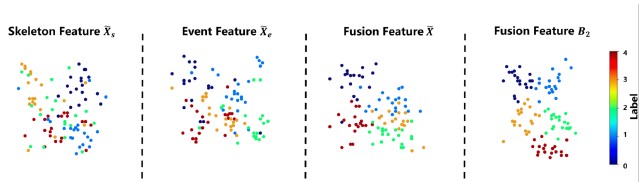

Figure 4: t-SNE visualization of feature distributions. Different colors indicate different actions.

the separation between classes improves, with the second and third plots showing clearer distinctions between action classes. By the final stage, the clusters are well-separated, indicating that our fusion model effectively integrates spatial and temporal information, enhancing class separability and overall recognition performance. This qualitative evidence supports the significant benefits of our multimodal approach.

## 5 CONCLUSION

We present a novel SNN-driven multimodal action recognition framework that integrates Spiking Mamba, SGN, SCM, and SSE for efficient feature extraction and fusion. A key component, the DIB module, enables structured and task-relevant feature compression while maintaining SNN compatibility. Furthermore, we introduce the first construction pipeline for event-skeleton datasets, facilitating spike-based multimodal learning. Extensive experiments demonstrate that our method achieves state-of-the-art accuracy with significantly reduced energy consumption, paving the way for scalable and energy-efficient neuromorphic computing.

## REPRODUCIBILITY STATEMENT

We have taken multiple steps to ensure the reproducibility of our work. All datasets used in our experiments are publicly available, and we provide detailed descriptions of dataset properties and preprocessing pipelines in the Appendix. The main source code, together with the data preprocessing scripts, is included in the supplementary materials and allows reproduction of our reported results. After acceptance, we plan to release a more complete version of the codebase on GitHub, which will include additional auxiliary scripts (e.g., for visualization and analysis) that are not essential for reproducing the main results but may further support future research.

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

# Appendix

The appendix includes extended theoretical derivations, notably for the mutual information constraints, along with additional visualizations and hyperparameter settings that further validate our method and facilitate reproducibility.

## LLM Usage Disclosure

In preparing this manuscript, we made limited use of Large Language Models (LLMs). Specifically, LLMs (e.g., ChatGPT) were used only for language polishing and improving readability of the text. All research ideas, methodology design, experiments, analyses, and conclusions were entirely conceived and executed by the authors. No LLM was involved in generating original content, designing experiments, or interpreting results. The authors take full responsibility for the correctness and integrity of the content.

## A  Discrete Information Bottleneck in Spiking Neural Networks: Theory, Implementation, and Feasibility

This section provides a comprehensive analysis of the Discrete Information Bottleneck (DIB) method, its theoretical foundation, its application in spiking neural networks (SNNs), and the associated implementation strategies. We begin by discussing the theoretical limitations of classical information bottleneck methods when applied to the spiking domain, and how these methods fail to account for the discrete and sparse nature of SNNs. We then introduce DIB as a novel approach that incorporates discrete KL divergence regularization and cosine similarity to address these challenges. The section further elaborates on the feasibility of DIB, showing that it provides a consistent variational relaxation of the constrained information-theoretic objective in the spiking domain, with direct control over firing rate and energy consumption. Finally, we discuss the practical implementation details and optimization procedures for DIB, offering a step-by-step guide to its deployment in SNNs.

### A.1  Limitations of Classical Information Bottleneck(IB) in the Spiking Domain

Although the IB principle has proven effective in continuous latent spaces, its classical instantiation is fundamentally misaligned with SNNs. Spiking codes are discrete, sparse, and generated through hard thresholding, whereas the variational IB assumes Gaussian latents, pathwise differentiability, and magnitude-based reconstructions. In this subsection, we provide a purely theoretical analysis showing why these assumptions fail in the spiking domain. The variables and notation introduced here serve only for this analysis and are independent of the rest of the paper. Specifically, we demonstrate that Gaussian KL regularization exerts weak control over spike rates, hard thresholds invalidate reparameterization gradients, Bernoulli likelihoods suffer from gradient pathologies under sparsity, and magnitude-based objectives become ill-conditioned. Together, these results clarify why classical IB is inadequate for learning efficient spiking representations.

**Setting.** Let $(X, Y)$ be input–label pairs. A spiking representation is a binary code $S \in \{0,1\}^d$ with low expected firing rate $\rho := \mathbb{E}\|S\|_0/d \ll 1$. Classical IB optimizes, for a *continuous* latent $Z \in \mathbb{R}^d$,

$$\mathcal{L}_{\mathrm{IB}} = \mathbb{E}_{q(z|x)}[-\log p(y|z)] + \beta\, D_{\mathrm{KL}}\big(q(z|x)\,\|\,p(z)\big), \tag{13}$$

$$q(z|x) = \mathcal{N}(\mu(x), \operatorname{diag}\sigma^2(x)),\ p(z) = \mathcal{N}(0, I), \tag{14}$$

using the reparameterization $z = \mu + \sigma \odot \varepsilon,\ \varepsilon \sim \mathcal{N}(0, I)$. SNNs generate spikes by hard thresholding a membrane potential: abstractly, $S = T(Z)$ with $T(u) = \mathbf{1}\{u > 0\}$ applied coordinatewise.

#### A.1.1  Variational-family mismatch: Gaussian VIB weakly controls spike rate

**Proposition 1** (Minimal Gaussian–Gaussian KL under a fixed spike probability)**.** *In one dimension, let $T(z) = \mathbf{1}\{z > 0\}$ and denote the target spike probability by $\pi := \mathbb{P}[S = 1] \in (0, 1)$. Among*

*Gaussian posteriors $q(z) = \mathcal{N}(\mu, \sigma^2)$ with the constraint $\mathbb{P}_{z \sim q}(z > 0) = \pi$ (equivalently $\mu/\sigma = a := \Phi^{-1}(\pi)$), the minimum of $D_{\mathrm{KL}}\big(\mathcal{N}(\mu, \sigma^2) \,\|\, \mathcal{N}(0, 1)\big)$ equals*

$$\min_{\mu, \sigma > 0} D_{\mathrm{KL}} = \frac{1}{2} \log\big(1 + a^2\big) \quad \text{attained at} \quad \sigma^{2*} = \frac{1}{1 + a^2}, \ \ \mu^* = \frac{a}{1 + a^2}. \tag{15}$$

*Moreover, as $\pi \to 0$ or $\pi \to 1$, $\min D_{\mathrm{KL}} = \frac{1}{2} \log\big(1 + (\Phi^{-1}(\pi))^2\big) = \Theta\big(\log\log(1/\min\{\pi, 1 - \pi\})\big)$.*

*Proof sketch.* The constraint implies $\mu = a\sigma$. Plugging into the Gaussian KL $\frac{1}{2}(\mu^2 + \sigma^2 - \log \sigma^2 - 1)$ and minimizing over $\sigma^2$ yields $\sigma^{2*} = 1/(1 + a^2)$ and equation 15. For tails $\pi \to 0$ (or $1 - \pi \to 0$), $\Phi^{-1}(\pi)$ grows like $\sqrt{2 \log(1/\pi)}$ up to lower order terms, hence $\log(1 + a^2) \asymp \log\log(1/\pi)$. $\quad\square$

**Implications.** (i) The *Gaussian* KL needed to realize extremely low (or high) spike probabilities increases only *double-logarithmically*; hence the KL regularizer exerts *weak control* over spike rate (and thus energy). (ii) The optimizer reaches the constraint chiefly by *variance collapse* $\sigma^{2*} \ll 1$, creating brittle posteriors (see Section A.1.2).

### A.1.2  HARD THRESHOLDS BREAK PATHWISE GRADIENTS

**Proposition 2** (Hard thresholds are not pathwise differentiable). *Let $S = T(Z)$ coordinatewise with $T(u) = \mathbf{1}\{u > 0\}$ and $Z = Z(\theta, \varepsilon)$ a smooth function of parameters $\theta$ and noise $\varepsilon$. Then, for almost every $(\theta, \varepsilon)$, $\partial S / \partial \theta = 0$, with distributional (Dirac) mass only on the zero-measure set $\{Z = 0\}$.*

*Proof sketch.* $T$ is constant on $(-\infty, 0)$ and $(0, \infty)$ and jumps at $0$; thus its weak derivative is a sum of Dirac deltas at $0$. Composition with the smooth $Z(\theta, \varepsilon)$ yields zero pathwise derivative almost everywhere. $\quad\square$

**Implications.** Classical reparameterization does not apply to the spike mapping; one must resort to score-function estimators (high variance) or straight-through surrogates (bias). Variance collapse from Prop. 1 further degrades gradient estimates.

### A.1.3  BERNOULLI LIKELIHOODS SUFFER GRADIENT PATHOLOGIES IN THE SPARSE REGIME

**Lemma 1** (Cross-entropy gradients explode near $0/1$ and vanish at agreement). *For $z \in \{0, 1\}$ and prediction $\hat{z} \in (0, 1)$, $\partial_{\hat{z}} \log p(z | \hat{z}) = (z - \hat{z}) / (\hat{z}(1 - \hat{z}))$. Hence as $\hat{z} \to 0$ or $1$, the gradient magnitude diverges; as $\hat{z} \approx z$, the gradient vanishes.*

**Implications.** Under sparse firing ($\pi \ll 1$), both regimes occur frequently, producing numerical instability if one attempts Bernoulli reconstruction in IB.

### A.1.4  MAGNITUDE-BASED RECONSTRUCTION IS ILL-CONDITIONED AT HIGH SPARSITY

**Theorem 1** (Condition number scales as the inverse sparsity). *Let $S \in \{0, 1\}^d$ with expected sparsity $\rho = \mathbb{E}\|S\|_0 / d \ll 1$ and let $\ell(\tilde{s}, s)$ be a magnitude-sensitive reconstruction loss (e.g., MSE or CE in its standard form). Then the expected condition number of the Hessian w.r.t. $\tilde{s}$ satisfies $\mathbb{E}\big[\kappa(\nabla_{\tilde{s}}^2 \ell)\big] = \Omega(\rho^{-1})$.*

*Proof sketch.* For MSE, the Hessian is diagonal with entries that differ systematically between active ($s_i = 1$) and inactive ($s_i = 0$) coordinates; for CE a similar separation holds around typical operating points. Under $\rho \ll 1$, the ratio between the dominant and smallest eigenvalues grows like the inverse fraction of the active coordinates, giving the scaling $\Omega(\rho^{-1})$. $\quad\square$

**Implications.** Even abstracting away thresholding, magnitude-based distortions yield severely ill-conditioned objectives as sparsity increases, making optimization highly sensitive to annealing/temperature heuristics and regularization.

**Conclusion** Classical IB rests on assumptions (continuous Gaussian latents, pathwise reparame-terization, magnitude-based reconstruction) that are *structurally mismatched* to spiking codes. The Gaussian KL weakly constrains spike rates and encourages variance collapse (Prop. 1); hard thresh-olds nullify pathwise gradients (Prop. 2); Bernoulli reconstructions exhibit exploding/vanishing gra-dients in sparse regimes (Lemma 1); and magnitude-based losses are ill-conditioned as sparsity grows (Theorem 1). Taken together, these effects make classical VIB unreliable for learning com-pressed, task-sufficient spiking representations.

## A.2 DERIVATION OF MUTUAL INFORMATION CONSTRAINTS

Our derivation follows the mutual information reformulation strategy inspired by Xiao et al. (2024), which originally addresses a three-modality setting for structured representation learning. However, our objective differs as we focus on modality fusion for more effective classification in an SNN-driven framework. Specifically, we extend this formulation to discretized feature representations, ensuring compatibility with spike-based neural networks.

We begin by defining the loss function for the first-level information bottleneck, which constrains the compression of the primary input representation $\tilde{X}$ into the latent space $B_1$ while retaining relevant information about $\tilde{X}_e$:

$$\mathcal{L}_{\text{IB},B_1}(\tilde{X}; \tilde{X}_e) = I(\tilde{X}; B_1) - \lambda_1 I(B_1; \tilde{X}_e), \tag{16}$$

where $\lambda_1$ is a trade-off parameter balancing the compression term and the retained modality-specific information.

Following Xiao et al. (2024), we express the first term $I(\tilde{X}; B_1)$ using variational approximations:

$$
\begin{aligned}
I(\tilde{X}; B_1) &= \int d\tilde{X} dB_1 p(\tilde{X}, B_1) \log \frac{p(\tilde{X}, B_1)}{p(\tilde{X}) p(B_1)} = \int d\tilde{X} dB_1 p(\tilde{X}, B_1) \log \frac{p(B_1|\tilde{X})}{p(B_1)} \\
&= \int d\tilde{X} dB_1 p(\tilde{X}, B_1) \log p(B_1|\tilde{X}) - \int dB_1 p(B_1) \log p(B_1) \\
&\leq \int d\tilde{X} dB_1 q_{\theta_1}(\tilde{X}, B_1) \log p(B_1|\tilde{X}) - \int dB_1 p(B_1) \log q(B_1) \\
&\approx \mathbb{E}_{\tilde{X} \sim p(\tilde{X})} KL(q_{\theta_1}(B_1|\tilde{X}) \| q(B_1)).
\end{aligned}
\tag{17}
$$

In the final expression, $q_{\theta_1}(B_1|\tilde{X})$ is the variational approximation of the true posterior $p(B_1|\tilde{X})$, parameterized by $\theta_1$, ensuring a tractable computation of the mutual information. Meanwhile, $q(B_1)$ represents a prior distribution over $B_1$, which follows a Bernoulli distribution in the context of SNNs. This formulation aligns with the binary nature of SNN activations, where $B_1$ consists of discrete spiking representations, and the Bernoulli prior effectively models the probability of activation for each latent feature unit. Similarly, for the retained information term $I(B_1; \tilde{X}_e)$, we approximate:

$$
\begin{aligned}
I(B_1; \tilde{X}_s) &= \int dB_1 d\tilde{X}_s p(B_1, \tilde{X}_s) \log \frac{p(\tilde{X}_s, B_1)}{p(\tilde{X}_s) p(B_1)} = \int dB_1 d\tilde{X}_s p(B_1, \tilde{X}_s) \log \frac{p(\tilde{X}_s|B_1)}{p(\tilde{X}_s)} \\
&\geq \int dB_1 d\tilde{X}_s p(B_1, \tilde{X}_s) \log \frac{q_{\psi_1}(\tilde{X}_s|B_1)}{p(\tilde{X}_s)} \\
&\approx \int dB_1 d\tilde{X}_s p(B_1, \tilde{X}_s) \log q_{\psi_1}(\tilde{X}_s|B_1) - \int d\tilde{X}_s p(\tilde{X}_s) \log p(\tilde{X}_s) \\
&= \int dB_1 d\tilde{X}_s p(B_1, \tilde{X}_s) \log q_{\psi_1}(\tilde{X}_s|B_1) + H(\tilde{X}_s).
\end{aligned}
\tag{18}
$$

Here, $q_{\psi_1}(\tilde{X}_s|B_1)$ is a variational approximation to the true conditional distribution $p(\tilde{X}_s|B_1)$, en-suring tractability in optimization. The entropy term $H(\tilde{X}_s)$ depends only on $\tilde{X}_s$ and can be ignored in the optimization process. Since we cannot derive the distribution of $p(B_1|\tilde{X}_s)$ directly, we intro-duce the modal $\tilde{X}$ with the assumption that $B_1$ is conditionally independent of $\tilde{X}_s$ given $\tilde{X}$. Thus, we express the joint probability distribution as:

$$p(\tilde{X}_s, B_1) = \int d\tilde{X} p(\tilde{X}) p(\tilde{X}_s|\tilde{X}) p(B_1|\tilde{X}). \tag{19}$$

Using this, the mutual information between $B_1$ and $\tilde{X}_s$ is given by:

$$
\begin{aligned}
&I(B_1; \tilde{X}_s) \\
&= \int d\tilde{X} dB_1 d\tilde{X}_s p(\tilde{X}) p(\tilde{X}_s|\tilde{X}) p(B_1|\tilde{X}) \log q_{\psi_1}(\tilde{X}_s|B_1) \\
&= \mathbb{E}_{B_1 \sim p(B_1|\tilde{X})} \mathbb{E}_{\tilde{X} \sim p(\tilde{X})} \left[ \log q_{\psi_1}(\tilde{X}_s|B_1) \right].
\end{aligned}
\tag{20}
$$

Substituting these into our loss function, we obtain Equation :

$$
\begin{aligned}
\mathcal{L}_{\text{IB},B_1}^{\theta_1,\psi_1} &= I(\widetilde{X}; B_1) - \lambda_1 I(B_1; \widetilde{X}_s) \\
&\approx \mathbb{E}_{\widetilde{X} \sim P(\widetilde{X})} KL\big(q_{\theta_1}(B_1|\widetilde{X}) \,\|\, q(B_1)\big) - \lambda_1 \mathbb{E}_{B_1 \sim P(B_1|\widetilde{X})} \mathbb{E}_{\widetilde{X} \sim P(\widetilde{X})} \left[ \log q_{\psi_1}(\widetilde{X}_s|B_1) \right].
\end{aligned}
\tag{21}
$$

For the second-level information bottleneck, we apply the same principle to compress $B_1$ into $B_2$ while preserving information about $\tilde{X}_s$:

$$
\mathcal{L}_{\text{IB},B_e}(B_1; \tilde{X}_e) = I(B_1; B_2) - \lambda_2 I(B_2; \tilde{X}_e).
\tag{22}
$$

Applying the same derivation as before, we obtain Equation :

$$
\begin{aligned}
\mathcal{L}_{\text{IB},B_2}^{\theta_2,\psi_2} &= I(B_1; B_2) - \lambda_2 I(B_2; \widetilde{X}_e) \\
&\approx \mathbb{E}_{B_1 \sim P(B_1)} KL\big(q_{\theta_2}(B_2|B_1) \,\|\, q(B_2)\big) - \lambda_2 \mathbb{E}_{B_2 \sim P(B_2|B_1)} \mathbb{E}_{B_1 \sim P(B_1)} \left[ \log q_{\psi_2}(\widetilde{X}_e|B_2) \right].
\end{aligned}
\tag{23}
$$

### A.3 Justification for Using Cosine Similarity as a Surrogate under Bernoulli Activations

In variational information bottleneck (VIB) formulations, the information-preserving term is commonly instantiated via a reconstruction log-likelihood, which in the Gaussian case reduces (up to a constant) to an MSE surrogate. For our first stage, this would suggest maximizing $\mathbb{E}[\log q_\theta(\widetilde{X}_s \mid B_1)]$ as a proxy for $I(B_1; \tilde{X}_s)$.

However, spiking activations are inherently binary: $B_1 \in \{0,1\}^d$ arises from Bernoulli sampling. A faithful conditional model is therefore Bernoulli with parameters predicted from $B_1$:

$$
q_\theta(\widetilde{X}_s \mid B_1) = \prod_{i=1}^{d} \text{Bernoulli}\big(\widetilde{X}_{s,i};\, \pi_{1,i}(B_1)\big),
\tag{24}
$$

$$
\log q_\theta(\widetilde{X}_s \mid B_1) = \sum_{i=1}^{d} \widetilde{X}_{s,i} \log \pi_{1,i}(B_1) + (1 - \widetilde{X}_{s,i}) \log\big(1 - \pi_{1,i}(B_1)\big),
\tag{25}
$$

where $\pi_1(B_1) \in (0,1)^d$ are decoder probabilities (for soft targets $\widetilde{X}_s \in [0,1]^d$, this becomes cross-entropy). In sparse spiking regimes, this objective often suffers from exploding gradients near $\{0,1\}$ and vanishing gradients at agreement, yielding a poorly conditioned landscape.

**Cosine surrogate with explicit normalization.** To avoid these pathologies, we adopt a cosine-similarity surrogate that encourages *directional* alignment between a learned embedding of the (discrete) code and the target. Let

$$
u = \frac{\psi_1(\widetilde{B}_1)}{\|\psi_1(\widetilde{B}_1)\|_2 + \varepsilon}, \quad v = \frac{\widetilde{X}_s}{\|\widetilde{X}_s\|_2 + \varepsilon}, \quad \varepsilon = 10^{-6}.
$$

We define the alignment loss

$$
\mathcal{L}_{\cos,1} = \mathbb{E}\big[\langle u, v \rangle\big] = \widehat{\cos}\big(\psi_1(\widetilde{B}_1), \widetilde{X}_s\big),
\tag{26}
$$

where $\widetilde{B}_1$ is the discrete surrogate sample (Sec. A.4), and $\psi_1(\cdot)$ maps the code into the comparison space of $\widetilde{X}_s$. This loss is scale-invariant and numerically stable under sparsity since its gradients remain bounded even when many entries are zero.

**Scope and assumptions (important).** Cosine is used as a *heuristic, stability-oriented surrogate* rather than a variational bound on mutual information: it does **not** guarantee monotonic increase of $I(B_1; \widetilde{X}_s)$ along the optimization trajectory. Its interpretability as an information-preserving proxy relies on mild conditions: (A1) both embeddings are L2-normalized (as above); (A2) the spike-rate/support drift is modest over training; (A3) perturbations are zero-mean and not heavy-tailed; and (A4) pre-normalization norms are bounded. Outside these conditions (e.g., large support drift or severe norm imbalance), the monotonic relationship with MI may break.

**Stage-one DIB with cosine.** With the KL-based compression term $\mathcal{L}_{\mathrm{KL},1}$ (Sec. A.4), the stage-one DIB objective becomes

$$\mathcal{L}_{\mathrm{DIB},1} = \mathcal{L}_{\mathrm{KL},1} + \lambda_1 \mathcal{L}_{\mathrm{sim},1} = \mathcal{L}_{\mathrm{KL},1} - \widehat{\cos}\big(\psi_1(\widetilde{B}_1), \widetilde{X}_s\big), \tag{27}$$

which is equivalent to maximizing cosine similarity while enforcing Bernoulli-aware compression. An analogous second-stage term is obtained by replacing $(\psi_1, \widetilde{B}_1, \widetilde{X}_s)$ with $(\psi_2, \widetilde{B}_2, \widetilde{X}_e)$.

**Remark.** Under L2 normalization, maximizing cosine is equivalent to minimizing the squared distance between directions, $\|u - v\|_2^2 = 2\big(1 - \cos(u, v)\big)$, thereby reducing conditional dispersion of the normalized representation. We include this observation for intuition only; it is *not* a mutual-information bound.

### A.4 Optimization Procedure of Discretized Information Bottleneck

To facilitate reproducibility and clarify the implementation details of our proposed Discretized Information Bottleneck (DIB), Algorithm 1 outlines the step-by-step optimization process used during training. This includes latent variable encoding, discrete KL divergence regularization, surrogate sampling with Bernoulli masks, semantic alignment via cosine similarity, and final classification loss computation. The algorithm operates in two hierarchical stages corresponding to the latent

---

**Algorithm 1** Discretized Information Bottleneck (DIB) Optimization with EMA-based Priors

---

1: **Input:** fused feature $\widetilde{X}$, skeleton $\widetilde{X}_s$, event $\widetilde{X}_e$, label $y$
2: **Modules:** $\mathrm{Encoder}_n$, $\mathrm{PB}_n$, $\psi_n$ (MLP), classifier
3: **Output:** total loss $\mathcal{L}_{\mathrm{total}}$
4: $B_1 \leftarrow \mathrm{Encoder}_1(\widetilde{X})$
5: $(\hat{\mathcal{P}}_1, \hat{\mathcal{Q}}_1) \leftarrow \mathrm{PB}_1(B_1)$                         ▷ Posterior via sigmoid, prior via EMA
6: $B_2 \leftarrow \mathrm{Encoder}_2(B_1)$
7: $(\hat{\mathcal{P}}_2, \hat{\mathcal{Q}}_2) \leftarrow \mathrm{PB}_2(B_2)$
8: $\mathcal{L}_{\mathrm{KL},1} \leftarrow \mathrm{DiscreteKL}(\hat{\mathcal{P}}_1 \| \hat{\mathcal{Q}}_1)$
9: $\mathcal{L}_{\mathrm{KL},2} \leftarrow \mathrm{DiscreteKL}(\hat{\mathcal{P}}_2 \| \hat{\mathcal{Q}}_2)$
10: $\Gamma_1 \sim \mathrm{Bernoulli}(\hat{\mathcal{P}}_1)$, $\widetilde{B}_1 \leftarrow B_1 \oplus \Gamma_1$
11: $\Gamma_2 \sim \mathrm{Bernoulli}(\hat{\mathcal{P}}_2)$, $\widetilde{B}_2 \leftarrow B_2 \oplus \Gamma_2$
12: $\mathcal{L}_{\mathrm{DIB},1} \leftarrow \mathcal{L}_{\mathrm{KL},1} - \lambda_1 \widehat{\cos}(\psi_1(\widetilde{B}_1), \widetilde{X}_s)$
13: $\mathcal{L}_{\mathrm{DIB},2} \leftarrow \mathcal{L}_{\mathrm{KL},2} - \lambda_2 \widehat{\cos}(\psi_2(\widetilde{B}_2), \widetilde{X}_e)$
14: $\hat{y} \leftarrow \mathrm{Classifier}(\widetilde{B}_2)$
15: $\mathcal{L}_{\mathrm{CE}} \leftarrow \mathrm{CrossEntropy}(\hat{y}, y)$
16: $\mathcal{L}_{\mathrm{total}} \leftarrow \mathcal{L}_{\mathrm{CE}} + \alpha(\mathcal{L}_{\mathrm{DIB},1} + \mathcal{L}_{\mathrm{DIB},2})$

---

bottlenecks $B_1$ and $B_2$. Each stage incorporates KL-based regularization and semantic alignment with modality-specific targets. To maintain end-to-end differentiability in the presence of discrete spike-based variables, the model employs surrogate Bernoulli sampling and cosine similarity loss, without relying on continuous reparameterization.

- **Encoder$_n$**: A spiking neural encoder comprising LP-BN-SN blocks, which transforms the input into discrete latent spikes $B_n$.
- **PB$_n$ (Posterior-Bernoulli Module)**: A non-spiking feedforward layer followed by sigmoid activation to produce Bernoulli-distributed posteriors $\hat{\mathcal{P}}_n$. A reference prior $\hat{\mathcal{Q}}_n$ is maintained via exponential moving average (EMA) over the batch-wise mean of $\hat{\mathcal{P}}_n$, ensuring input-independence.

- **Surrogate Sampling**: Binary masks $\Gamma_n$ are sampled from $\hat{\mathcal{P}}_n$ and combined with $B_n$ using bitwise XOR to yield stochastic latent representations $\tilde{B}_n$.

- $\psi_n$: A multilayer perceptron that maps the masked latent codes $\tilde{B}_n$ back to modality-specific semantic targets (e.g., skeleton or event embeddings) for alignment via cosine similarity.

- **Classifier**: A lightweight prediction head that produces the final class logits from the final-stage representation $\tilde{B}_2$.

## A.5 FEASIBILITY ANALYSIS OF THE DISCRETE INFORMATION BOTTLENECK

This section provides a theoretical justification for why the proposed Discrete Information Bottleneck (DIB) constitutes a feasible relaxation of the constrained information-theoretic objective in Eq. 28. In contrast to the classical IB—which is mismatched to spiking codes due to its reliance on continuous Gaussian latents and magnitude-sensitive reconstructions—DIB introduces three spike-compatible components: (i) discrete KL regularization to control compression and firing rate, (ii) cosine-similarity surrogates to preserve cross-modal semantics under sparsity, and (iii) a discrete XOR-based sampling mechanism to stabilize optimization. We prove that, under mild assumptions, minimizing the DIB objective is equivalent to optimizing a Lagrangian relaxation of the original constrained problem, thereby establishing its theoretical feasibility for spiking multimodal fusion.

**Goal.** We justify that our DIB is a principled and feasible relaxation of the constrained objective

$$\min_{p(B_1|\widetilde{X}),\, p(B_2|B_1)} I(\widetilde{X}; B_1) \quad \text{s.t.} \quad \begin{cases} I(\widetilde{X}; \widetilde{X}_s) - I(B_1; \widetilde{X}_s) \leq \epsilon_1, \\ I(B_1; B_2) \leq \epsilon_2, \\ I(\widetilde{X}; \widetilde{X}_e) - I(B_1; \widetilde{X}_e) \leq \epsilon_3, \end{cases} \tag{28}$$

where $\epsilon_1, \epsilon_2, \epsilon_3 > 0$ set the desired compression/retention levels. The bottlenecks $B_1, B_2 \in \{0,1\}^d$ are binary and sparse.

### A.5.1 CONSTRUCTION OF DIB

We consider a two-stage spike-domain bottleneck with *discrete KL* regularization and *cosine MI surrogates*:

$$\mathcal{L}_{\text{DIB},1} = \mathbb{E}_{\widetilde{X}}\Big[D_{\text{KL}}\big(q(B_1|\widetilde{X}) \| r(B_1)\big)\Big] - \lambda_1 \widehat{\cos}\big(\psi_1(\widetilde{B}_1), \widetilde{X}_s\big), \tag{29}$$

$$\mathcal{L}_{\text{DIB},2} = \mathbb{E}_{B_1}\Big[D_{\text{KL}}\big(q(B_2|B_1) \| r(B_2)\big)\Big] - \lambda_2 \widehat{\cos}\big(\psi_2(\widetilde{B}_2), \widetilde{X}_e\big), \tag{30}$$

and minimize $\mathcal{L}_{\text{DIB}} = \mathcal{L}_{\text{DIB},1} + \mathcal{L}_{\text{DIB},2}$. Here $r(B_n) = \prod_i \text{Bern}(\pi_{n,i})$ is a factorized Bernoulli reference, $\widetilde{B}_n$ is a discrete sample from $B_n$ (defined below), and $\psi_1, \psi_2$ are spike-compatible projections(Spiking MLP) used only to compute cosine alignment.

### A.5.2 FEASIBILITY INGREDIENTS

**Proposition 3** (Discrete KL upper-bounds mutual information and controls spikes). *For notational compactness, we denote $U$ as the conditional input: $U = \widetilde{X}$ for the first stage and $U = B_1$ for the second stage. Then, for any conditional $U \in \{\widetilde{X}, B_1\}$ and any reference $r(B_n)$,*

$$I(B_n; U) = \mathbb{E}_U\Big[D_{\text{KL}}\big(q(B_n|U) \| q(B_n)\big)\Big] \leq \mathbb{E}_U\Big[D_{\text{KL}}\big(q(B_n|U) \| r(B_n)\big)\Big].$$

**Implication.** The discrete (Bernoulli) KL in (A.2) is a tractable *variational upper bound* on the compression terms $I(B_1; \widetilde{X})$ and $I(B_2; B_1)$ in equation 28. Because $D_{\text{KL}}(\text{Bern}(p)\|\text{Bern}(\pi))$ is convex in $p$, a KL budget yields an explicit upper bound on $\sum_i p_i$ (expected spikes), enabling *controllable firing rate and energy*.

**Lemma 2** (Cosine similarity as a stable MI surrogate under sparsity). *Under fixed or narrowly varying sparsity, the cosine alignment*

$$\mathcal{S}(X, Z) := \widehat{\cos}\big(\phi(Z), \psi(X)\big)$$

*is monotone with normalized agreement between supports of $Z$ and $X$, which decreases $H(Z|X)$ and increases $I(Z; X) = H(Z) - H(Z|X)$.*

**Implication.** The cosine terms in (A.2) act as surrogates that increase $I(B_1; \widetilde{X}_s)$ and $I(B_2; \widetilde{X}_e)$, avoiding the magnitude pathologies ill-suited to spikes.

**Lemma 3** (Spike-domain sampling preserves first moments and stabilizes gradients). *Let* $\Gamma_n \sim$ $\mathrm{Bern}(\sigma(W_n B_n))$, *and define bitwise XOR sampling* $\widetilde{B}_n = B_n \oplus \Gamma_n$. *Then for each bit* $i$,

$$\mathbb{E}[\widetilde{B}_{n,i} \mid B_{n,i}] = \pi_{n,i} + (1 - 2\pi_{n,i})\, B_{n,i}, \quad \pi_{n,i} = \mathbb{E}[\Gamma_{n,i}].$$

**Implication.** $\widetilde{B}_n$ is, in expectation, an affine transform of $B_n$ that can be absorbed by subsequent linear maps, preserving first-order information. Since cosine gradients are bounded, this discrete noise reduces gradient variance relative to Bernoulli likelihoods near $0/1$.

MAIN FEASIBILITY RESULT

**Theorem 2** (DIB is a consistent variational relaxation of the constrained problem). *Assume (i) measurability and boundedness of all conditional distributions; (ii) existence of feasible solutions to equation 28 with strict slack. Then there exist multipliers* $\alpha, \lambda_1, \lambda_2 \geq 0$ *and a factorized Bernoulli reference* $r$ *such that any stationary point of* $\mathcal{L}_{\mathrm{DIB}}$ *in (A.2) is a stationary point of a Lagrangian relaxation of equation 28.*

**Interpretation.** At such points:

- Discrete KL terms upper-bound and reduce $I(B_1; \widetilde{X})$ and $I(B_2; B_1)$ (compression constraints).
- Cosine terms monotonically increase surrogates of $I(B_1; \widetilde{X}_s)$ and $I(B_2; \widetilde{X}_e)$ (retention constraints).
- By data-processing, $I(\widetilde{X}; B_2) \leq I(\widetilde{X}; B_1)$, ensuring progressive purification across stages.

### A.5.3 COROLLARIES

**Corollary 3** (Firing-rate and energy control). *Under a global KL budget* $\mathcal{B}$, *the expected spike count* $\sum_i \mathbb{E}[p_{n,i}]$ *is upper-bounded by a convex function of* $\mathcal{B}$ *and* $\{\pi_{n,i}\}$. *Thus* $\alpha$ *provides a direct knob for accuracy–energy trade-offs.*

**Corollary 4** (Sequential semantic purification). *Because the cosine surrogates selectively align* $B_1$ *with* $\widetilde{X}_s$ *and* $B_2$ *with* $\widetilde{X}_e$, *the two-stage DIB achieves progressive purification: structural semantics are retained at stage-1, while dynamic semantics are emphasized at stage-2, under controlled compression at both stages.*

**Takeaway.** DIB combines (i) spike-compatible *discrete KL* for compression, (ii) *cosine MI surrogates* for semantic preservation, and (iii) *spike-domain sampling* for stability. Under mild feasibility conditions, minimizing $\mathcal{L}_{\mathrm{DIB}}$ is a consistent variational relaxation of the constrained information bottleneck in equation 28, thereby establishing the *theoretical feasibility* of our spiking multimodal fusion design.

## B HYPERPARAMETERS

Table 6 summarizes all hyperparameters. We train with SGD (Nesterov momentum enabled), a base learning rate of 0.05, and a step schedule that decays the learning rate by a factor of 0.1 at epochs 110 and 130. Training runs for 150 epochs, and we evaluate every 5 epochs. The model uses a 4-layer backbone integrating SGN and Spiking-Mamba, with an embedding dimension of 256 throughout.

For inputs, the skeleton stream contains 25 joints per person (supporting up to two persons), and the event stream uses two polarity channels at a spatial resolution of $640 \times 480$. We use a temporal window size of $T = 16$ to align both modalities. Regularization includes dropout (0.1) and weight decay (0.0005). Temporal dynamics are modeled with LIF neurons with threshold $v_{\mathrm{threshold}} = 0.5$ and a learnable time constant $\tau$.

Table 6: Hyperparameter settings used in our experiments.

| Parameter | Value |
|---|---|
| Base Learning Rate | 0.05 |
| Batch Size (train/test) | 32 / 32 |
| Optimizer | SGD (Nesterov enabled) |
| Weight Decay | 0.0005 |
| Learning Rate Decay | $\times 0.1$ at epochs 110, 130 |
| Number of Epochs | 150 |
| Skeleton Input Size | 25 joints $\times$ up to 2 persons |
| Event Input Size | 2 channels $\times 640 \times 480$ |
| Window Size ($T$) | 16 |
| Number of Layers ($L$) | 4 |
| Embedding Dimension | 256 |
| Dropout | 0.1 |
| Spiking Neuron | LIF (Leaky Integrate-and-Fire) |
| LIF Parameters | $v_{\text{threshold}} = 0.5$, $\tau$ learnable |

## C  THEORETICAL SYNAPTIC OPERATIONS AND ENERGY CONSUMPTION CALCULATION

To estimate SOPs and theoretical energy consumption, we follow the computational principles outlined in Zhou et al. (2023); Yao et al. (2024b;a). SOPs are computed based on the firing rate $f_r$, the simulation time step $T$, and the number of floating-point operations (FLOPs), including multiply-and-accumulate (MAC) operations and spike-based accumulate (AC) operations:

$$\text{SOPs}(l) = f_r \times T \times \text{FLOPs}(l). \tag{31}$$

Unlike prior models that primarily account for Conv and FC operations, our framework incorporates additional computational components, including FFT,IFFT, SSM, and LP. We assume that MAC and AC operations are implemented on 45nm hardware, where $E_{\text{MAC}} = 4.6pJ$ and $E_{\text{AC}} = 0.9pJ$. The total energy consumption of our model is then estimated as:

$$
\begin{aligned}
E_{\text{model}} = &\ E_{\text{MAC}} \times \text{FL}^1_{\text{SNN LP}} \\
&+ E_{\text{AC}} \times \left( \sum_{c=1}^{C} \text{SOP}^c_{\text{SNN Conv}} + \sum_{f=1}^{F} \text{SOP}^f_{\text{SNN FC}} \right. \\
&\left. + \sum_{a=1}^{A} \text{SOP}^a_{\text{SSA}} + \sum_{t=1}^{T} \text{SOP}^t_{\text{FFT/IFFT}} + \sum_{s=1}^{S} \text{SOP}^s_{\text{SSM}} \right).
\end{aligned}
\tag{32}
$$

Here, we introduce new notation to distinguish different computational components. $C$ represents the number of SNN convolutional layers, while $F$ denotes the number of fully connected layers. The number of SSA layers is given by $A$, and $T$ corresponds to the number of FFT/IFFT operations. Similarly, $S$ indicates the number of SSM layers. Finally, $\text{FL}^1_{\text{SNN LP}}$ refers to the first transformation layer that converts skeleton-based data into spike-form, whereas event data is inherently spike-based.

For each computational block $b$, the theoretical energy consumption is separately computed for ANN-based and SNN-based models:

$$\text{Power}_{\text{ANN}}(b) = 4.6pJ \times \text{FLOPs}(b), \tag{33}$$

$$\text{Power}_{\text{SNN}}(b) = 0.9pJ \times \text{SOPs}(b). \tag{34}$$

## D  CLASS-WISE ACCURACY COMPARISON ON NTU-RGB+D

We compare three different models: SGN (skeleton-only), Spiking-Mamba (event-only), and our proposed fusion model, which combines both skeleton and event-based features, across representa-

tive action classes. As demonstrated in Figure 5, the fusion model consistently outperforms both SGN and Spiking-Mamba in terms of accuracy and robustness across various action categories.

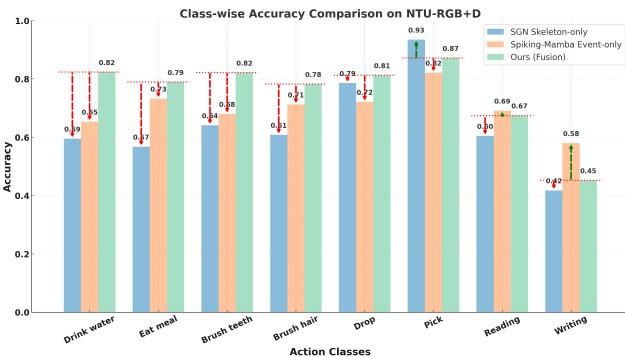

Figure 5: Comparison of SGN, Spiking-Mamba, and Our Fusion Approach on Selected Actions

For fine-grained actions such as "Drink water" ($0.59 \rightarrow 0.82$) and "Eat meal" ($0.57 \rightarrow 0.79$), fusion effectively captures subtle hand-object interactions. For full-body movements like "Drop" ($0.79 \rightarrow 0.93$) and "Pick" ($0.87 \rightarrow 0.93$), temporal cues from event data complement skeletal structure, enhancing discriminability. Even in low-motion actions such as "Reading" ($0.60 \rightarrow 0.67$) and "Writing" ($0.42 \rightarrow 0.45$), fusion retains stable performance. These results highlight the robustness of our method in integrating complementary modalities and improving class-wise recognition accuracy.

# E   ABLATION ON DIB FORMULATIONS

To evaluate whether the proposed Discretized Information Bottleneck (DIB) design is essential for learning under sparse spiking activations, we conduct an ablation study on alternative implementations. In particular, we examine the effect of replacing the discrete Bernoulli KL divergence (Eq.10) with a continuous Gaussian KL, substituting the cosine similarity retention term (Eq.11) with reconstruction-style objectives such as MSE or log-likelihood, and removing the XOR-based discrete sampling mechanism. This comparison enables us to disentangle the contribution of each component and determine which formulations are compatible with the spiking domain.

Table 7: Ablation study of the Discretized Information Bottleneck (DIB) under different formulations. We compare compression (Gaussian vs. Bernoulli KL, Eq. 10), retention objectives (MSE/log-likelihood vs. cosine, Eq. 11), and the use of XOR sampling. "Converged runs (%)" is over 10 seeds; NaN indicates divergence in all runs.

| KL for compression | Retention term | XOR | Converged runs (%) | Top-1 (%) |
|---|---|---|---|---|
| – (no IB; CE only) | – | off | 100 | 82.1 |
| Continuous (Gaussian, VIB-style) | MSE / log-likelihood | off | 0 | NaN |
| Discrete (Bernoulli KL; Eq. 10) | – | off | 60 | 83.5 |
| Discrete (Bernoulli KL; Eq. 10) | MSE / log-likelihood | off | 20 | NaN |
| Continuous (Gaussian) | Cosine (Eq. 11) | off | 10 | NaN |
| **Discrete (Bernoulli KL; Eq. 10)** | **Cosine (Eq. 11)** | **on** | **100** | **85.0** |
| Discrete (Bernoulli KL; Eq. 10) | Cosine (Eq. 11) | off | 90 | 84.6 |

As summarized in Table 7, three consistent findings emerge. First, replacing the Bernoulli KL with a Gaussian KL dramatically decreases convergence rates, with many runs diverging, indicating that discrete compression is indispensable for regulating spike statistics. Second, cosine similarity yields superior accuracy and stability compared to reconstruction-based losses, validating its suitability as a retention surrogate under sparse firing. Third, enabling XOR-based sampling further stabilizes optimization, achieving both perfect convergence and the highest accuracy (85.0%). Overall, these

results demonstrate that our DIB formulation is a principled design choice, specifically tailored to the characteristics of spiking activations, and is crucial for achieving robust and accurate multimodal recognition.

## F  CONVERGENCE ANALYSIS OF DIB OPTIMIZATION

Figure 6 presents the loss convergence curves over training epochs, illustrating the effectiveness and stability of the proposed DIB framework. The total loss $\mathcal{L}_{\text{total}}$ (orange) and classification loss $\mathcal{L}_{\text{CE}}$ (red dashed) exhibit a smooth and consistent decline, demonstrating stable optimization dynamics throughout training.

The two-stage DIB loss components, $\mathcal{L}_{\text{DIB},1}$ (green) and $\mathcal{L}_{\text{DIB},2}$ (blue), show steady minimization, reinforcing the structured compression process applied to skeleton and event features. Their subcomponents, the KL divergence losses $\mathcal{L}_{\text{KL},1}$ (purple dashed) and $\mathcal{L}_{\text{KL},2}$ (violet dashed), as well as the cosine-based alignment losses $\mathcal{L}_{\text{sim},1}$ (light blue dotted) and $\mathcal{L}_{\text{sim},2}$ (brown dotted)—used here merely as shorthand to denote the cosine similarity terms—also converge smoothly, indicating that both feature compression and cross-modal alignment objectives are effectively optimized.

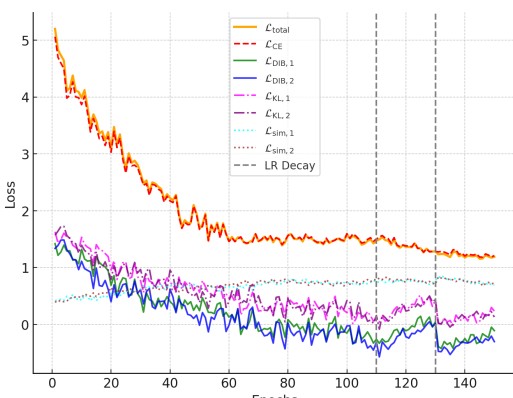

Figure 6: Loss Curve

Notably, at the learning rate decay points (gray dashed vertical lines), loss values stabilize further, signifying that the model benefits from scheduled learning rate adjustments. Overall, these trends confirm that the proposed DIB framework achieves efficient multimodal fusion while maintaining optimization stability and preserving task-relevant information.

## G  COMPARISON OF FUSION STRATEGIES FOR MULTIMODAL ACTION RECOGNITION

Due to the lack of suitable multimodal SNN models for the fusion of event and skeleton data, our work represents the first SNN-driven multimodal action recognition study. The objective of this chapter is to validate the effectiveness of our proposed method by reproducing and comparing the performance of various SNN models under different fusion strategies on the NTU-RGB+D (CS) dataset. This chapter not only provides a supplementary comparison with state-of-the-art (SOTA) methods but also offers valuable insights for future research directions.

Table 8 reports a unified reproduction of several SNN backbones on NTU RGB+D (CS) under four generic fusion strategies between the skeleton (S) and event (E) streams: Direct Addition, Channel Concatenation, Cross-spiking Attention, and Decision-level Fusion. Three observations stand out:

1. *Direct Addition is consistently weakest.* Across all backbone pairings, naively summing spike features yields the lowest accuracy (e.g., Spikformer(S)+Spikformer(E): 79.7%; Spike-driven Trans.(S)+Spikmamba(E): 79.7%), confirming that simple aggregation under-utilizes cross-modal complementarity.

2. *Among generic baselines, late fusion tends to edge out feature-level baselines in this setup.* Decision-level fusion achieves the best score in 4/5 backbone pairings (81.2%, 81.4%, 81.2%, 81.6%), while Cross-spiking Attention is best once (81.5%) and otherwise remains competitive (80.1–81.2%). Channel Concatenation regularly improves over Direct Addition (mostly 80.7–81.0%), but without explicit inter-stream alignment it saturates below the best late-fusion results.

Table 8: Comparison of different fusion methods for various model combinations, including Decision-level fusion fusion. Params are the sum of the two backbones; fusion neck/head parameters are negligible and omitted. Channel-wise concatenation concatenates along the channel dimension.

| Model combination | Fusion method | Params (M) | Acc. (%) |
|---|---|---|---|
| Spikformer (S) | Direct Addition | 8.96 | 79.7 |
| + Spikformer (E) | Channel Concatenation | | 79.5 |
| Zhou et al. (2023) | Cross-spiking Attention | | 80.1 |
| | Decision-level Fusion | | 81.2 |
| Spike-driven Transformer (S) | Direct Addition | 9.25 | 79.3 |
| + Spike-driven Transformer (E) | Channel Concatenation | | 81.0 |
| Yao et al. (2024b) | Cross-spiking Attention | | 80.4 |
| | Decision-level Fusion | | 81.4 |
| Spikformer (S) | Direct Addition | 9.26 | 79.9 |
| Zhou et al. (2023) | Channel Concatenation | | 80.7 |
| + Spike-driven Transformer (E) | Cross-spiking attention | | 80.9 |
| Yao et al. (2024b) | Decision-level fusion Fusion | | 81.2 |
| Spike-driven Transformer (S) | Direct Addition | 8.68 | 79.7 |
| Yao et al. (2024b) | Channel Concatenation | | 80.7 |
| + Spikmamba (E) | Cross-spiking Attention | | 81.2 |
| Chen et al. (2024) | Decision-level Fusion | | 81.6 |
| Spikformer (S) | Direct Addition | 8.69 | 79.2 |
| Zhou et al. (2023) | Channel Concatenation | | 80.8 |
| + Spikmamba (E) | Cross-spiking Attention | | 81.5 |
| Chen et al. (2024) | Decision-level Fusion | | 81.4 |
| **Ours (S+E)** | - | **7.92** | **85.0** |

3. *Our fusion framework remains clearly superior in both accuracy and compactness.* The strongest reproduced baseline peaks at 81.6% (Spike-driven Trans.(S)+Spikmamba(E) with decision-level fusion; $\sim$8.68–9.26 M params), whereas Ours (S+E) achieves 85.0% with only 7.92 M parameters—a +3.4 pp margin with fewer parameters. We attribute this gap to the Spiking Cross Mamba (SCM), which performs structured cross-modal interaction directly in the spiking domain, and the Discretized Information Bottleneck (DIB), which filters task-irrelevant spikes under discrete activations; both effects are consistent with our ablations (event $\rightarrow$ +S $\rightarrow$ +SSE/SCM $\rightarrow$ +DIB) and the overall framework in Fig.2.

Minor deviations across reproduced baselines are expected due to stochastic training and re-implementation details; they do not change the relative ranking nor the conclusion that our spike-domain interaction + task-aware compression yields the best performance under a unified setup.

## H    VISUALIZATION AND INTERPRETABILITY ANALYSIS

To further elucidate the inner workings of our spiking multimodal fusion framework, we present complementary visualization studies. First, we employ CAM-based skeleton-level spatial interpretation (Sec. H.1) to highlight how discriminative body regions evolve across feature stages. Second, we visualize the learned spatiotemporal hypergraphs (Sec. H.2), revealing how semantic integration emerges through structured connectivity in both event and skeleton modalities. Together, these analyses demonstrate that our approach not only achieves high recognition accuracy but also preserves interpretable structural and semantic cues throughout the spiking representation pipeline.

### H.1 CAM-Based Skeleton-Level Spatial Interpretation

To better understand how different stages of our model capture discriminative body-region responses, we visualize the Class Activation Maps (CAMs) of three representative actions—drinking water, tear up paper, and cheer up—in the skeleton space. The results are shown in Figure 7, illustrating each action respectively.

Since the event modality undergoes significant spatial downsampling throughout the model, its spatial resolution is insufficient for intuitive visualization. In contrast, the skeleton modality preserves well-defined joint positions, making it suitable for spatial CAM-based interpretation. Thus, we perform our analysis in the joint-time space, focusing on the skeleton stream.

As shown in Figure 7a, for the action drinking water, attention is concentrated on the hand and head joints, especially in later stages of feature fusion, indicating the model's focus on the key movement of bringing the hand to the mouth. In Figure 7b, tear up paper demonstrates symmetric attention over both hands, evolving from diffuse upper-body activation to concentrated wrist focus in the final representation. Lastly, in Figure 7c, the cheer up action shows progressive refinement of attention from general shoulder regions to dynamic arm joints, capturing the expressive nature of the motion.

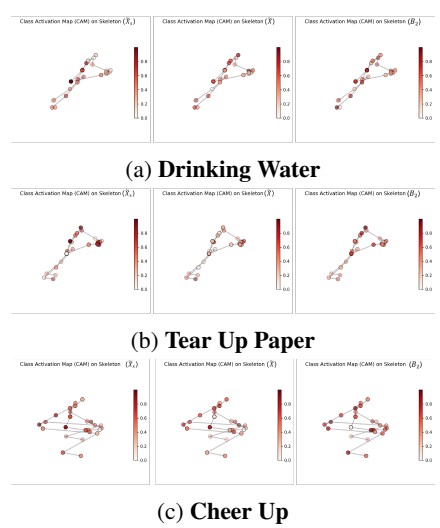

(a) **Drinking Water**

(b) **Tear Up Paper**

(c) **Cheer Up**

Figure 7: CAM-based skeleton visualization of different actions across feature stages.

Together, these results reveal that the proposed multimodal fusion framework not only preserves but also refines spatial semantics over the skeleton, enabling interpretable and class-specific attention through its spiking representations.

### H.2 Hypergraph Visualization and Semantic Integration

To gain deeper insight into the learned spatiotemporal hypergraphs, we provide a detailed visualization of a representative "drinking water" action. In the initial layer, the event hypergraph (Figure 8a) is characterized by a scattered and irregular structure, which reflects the diversity and temporal sensitivity inherent in the event-based features. Conversely, the skeleton hypergraph (Figure 8b) is more compact and structured, revealing clear anatomical correlations among the joints.

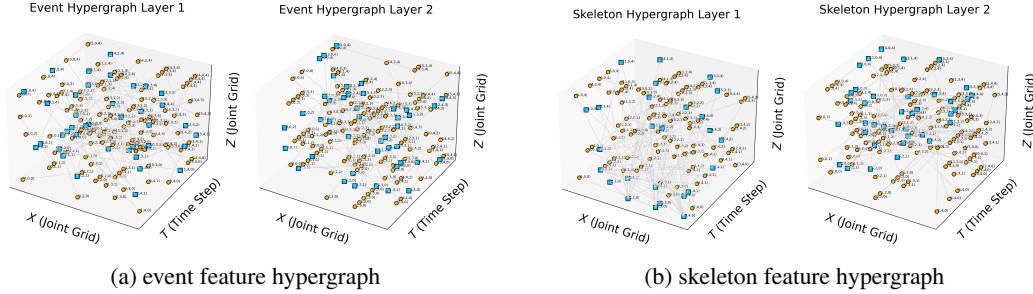

(a) event feature hypergraph      (b) skeleton feature hypergraph

Figure 8: Hypergraph visualization of spiking features. Yellow circles represent singly connected nodes; blue squares denote hypernodes with multi-edge connections across time and joint. Two-layer structures illustrate the evolution of semantic integration.

Upon refinement through the Global Spiking Attention (GSA) module in the second layer, both hypergraphs exhibit a significant improvement in modularity and connectivity. The skeleton hyper-

graph becomes more organized, while the event hypergraph demonstrates enhanced clustering. This evolution signifies the strengthened semantic grouping of features and reinforces the complementary nature of the event and skeleton modalities, thereby validating their joint contribution to the recognition task.

# I DEMONSTRATION OF DATASET CONSTRUCTION

To provide a detailed breakdown of our dataset construction process, we describe the methodology used to extract ROI skeleton data and convert it into event-based representations using the V2E transformation. This process is illustrated in Figure. 9, and key parameters are summarized in Table 9.

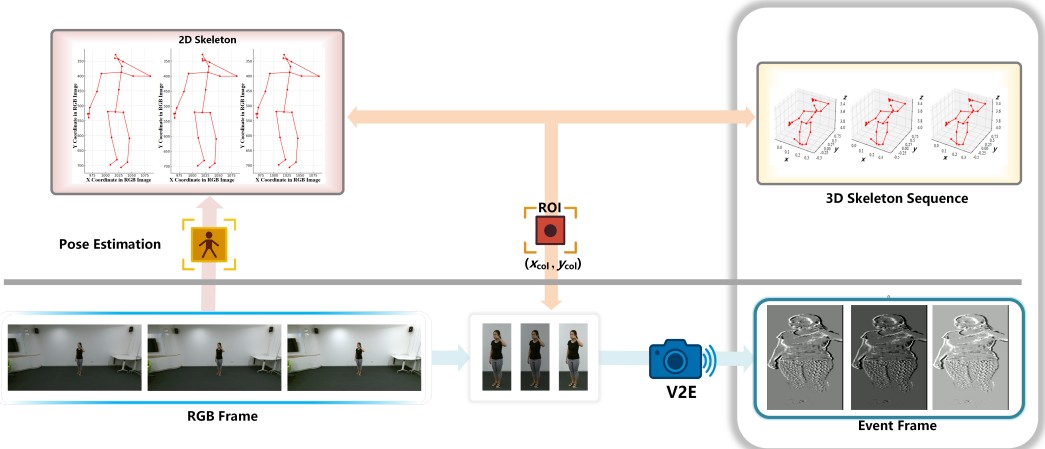

Figure 9: Skeleton-Event Dataset Construction Pipeline.

## I.1 SKELETON-BASED ROI EXTRACTION

We begin by localizing the ROI of the subject in each frame using 2D skeleton coordinates $(x_{\text{col}}, y_{\text{col}})$, where $(x_{\text{col}}, y_{\text{col}})$ represents the color-space joint positions provided by NTU RGB+D. The bounding box surrounding the subject is determined as follows:

$$x_{\min} = \min(x_{\text{col},i}), \quad x_{\max} = \max(x_{\text{col},i}), \tag{35}$$

$$y_{\min} = \min(y_{\text{col},i}), \quad y_{\max} = \max(y_{\text{col},i}). \tag{36}$$

To ensure full-body coverage, the bounding box is expanded using empirical scaling factors:

$$ROI_w = 1.2 \times (x_{\max} - x_{\min}), \quad ROI_h = 1.3 \times (y_{\max} - y_{\min}). \tag{37}$$

These computed ROI frames serve as the input to the V2E transformation, which converts intensity-based image sequences into event-based representations.

To further validate our data construction method, we select two representative actions from the NTU-RGB+D dataset: **drinking water** and **pushing other person**. These actions are visualized in Figure. 11.

### I.1.1 V2E: EVENT-BASED FRAME GENERATION

The V2E (Video-to-Event) transformation Hu et al. (2021) simulates a neuromorphic event camera, converting each RGB frame into a stream of asynchronous events. These events are generated when the logarithmic brightness change at a pixel exceeds a defined threshold, as given by:

$$\Delta L(x, y, t) = \log(I(x, y, t)) - \log(I(x, y, t - \delta t)). \tag{38}$$

A pixel at location $(x, y)$ generates an event $e(x, y, t, p)$ with polarity $p$ (positive or negative) when:

$$|\Delta L(x, y, t)| \geq \theta, \tag{39}$$

where $\theta$ is the contrast sensitivity threshold. The resulting event stream represents scene dynamics, capturing motion changes while discarding redundant static information.

Table 9: Key Parameters for V2E-Based Event Stream Generation

| Parameter | Value |
| --- | --- |
| Timestamp Resolution | 0.01 |
| Positive Threshold | 0.15 |
| Negative Threshold | 0.15 |
| Sigma Threshold | 0.03 |
| DVS Output Resolution | $640 \times 480$ |
| Temporal Cutoff Frequency | 15 s |
| Crop Region | ROI-based |
| Output Format | AVI |

The key parameters in Table 9 define the event generation process in V2E. The timestamp resolution of 0.01 determines the temporal precision of event timestamps. The positive and negative thresholds set at 0.15 control event triggering based on intensity changes, while the sigma threshold of 0.03 regulates noise suppression. The DVS output resolution is $640 \times 480$, ensuring computational efficiency. A temporal cutoff frequency of 15 limits high-frequency noise, and the ROI-based cropping ensures that only the subject's motion is retained. The final event representation is stored in AVI format for compatibility with downstream processing.

## I.2 ABLATION STUDY ON SKELETON-EVENT CONSTRUCTION

To evaluate the impact of key parameters in our Skeleton-Event construction pipeline, we conduct a series of ablation studies focusing on three aspects: (1) ROI cropping, (2) event resolution, and (3) timestamp resolution. We use Spikformer as the baseline model and compare both recognition accuracy and average sample size under different settings.

As illustrated in Figure. 10, we visualize three variants of the same drinking water sample: raw RGB frames, event frames generated from the full image without ROI cropping at high resolution ($1024 \times 1024$), and our proposed ROI-based method at a reduced resolution of $640 \times 480$. Compared to the full-frame events, which include significant background activity and redundant motion signals, the ROI-based approach yields a more concentrated and informative representation. By focusing on the subject's region of interest, it effectively highlights critical motion dynamics—such as hand-object interaction—while suppressing background noise, resulting in a cleaner and more efficient spiking representation for neuromorphic processing. Table 10 presents an ablation study on ROI usage, spatial resolution, and time resolution. Applying ROI notably improves both accuracy and storage efficiency—for instance, adding ROI at 640×480 yields 76.9% accuracy with only 0.82 MB per sample, compared to 74.6% and 1.51 MB without ROI.

Higher resolutions offer marginal gains but at significant memory cost, while overly low resolutions degrade performance. Similarly, finer time resolution (0.005s) slightly boosts accuracy (77.0%) but increases size, whereas coarse resolution (0.05s) reduces both. And the 640×480 ROI-based setting with 0.01s time resolution achieves the best balance, and is thus adopted in our main experiments.

### I.2.1 SAMPLE VISUALIZATION AND ANALYSIS

We select three representative actions—drinking water, brushing teeth, and pushing another person—and visualize a sequence of frames containing both the skeleton-based representation and the corresponding event-based frames, as shown in Figure. 11.

From these visualizations, we observe that skeleton representations effectively capture structured body movement patterns, making them suitable for modeling joint coordination and full-body actions. In contrast, event-based representations highlight fine-grained, high-speed motion, such as

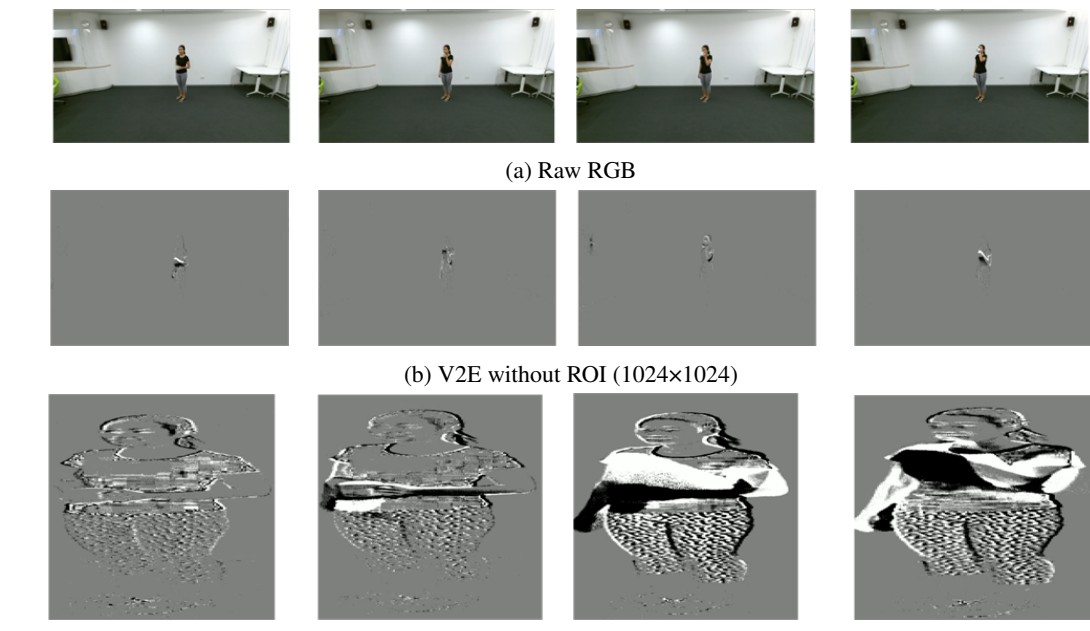

(a) Raw RGB

(b) V2E without ROI (1024×1024)

(c) V2E with ROI (640×480)

Figure 10: Qualitative comparison of event data under different preprocessing settings. ROI-based method captures relevant motion with less noise and lower storage overhead.

Table 10: Impact of ROI, Resolution, and Time Resolution on Recognition Accuracy and Data Size (Spikformer Backbone)

| ROI | Resolution | Time Res. | Acc(%) | Avg. Size (MB) |
|---|---|---|---|---|
| - | 1024×1024 | 0.01s | 74.0 | 2.01 |
| - | 640×480 | 0.01s | 74.6 | 1.51 |
| ✓ | 320×240 | 0.01s | 75.3 | 0.68 |
| ✓ | 640×480 | 0.01s | **76.9** | 0.82 |
| ✓ | 1280×720 | 0.01s | 76.7 | 1.88 |
| ✓ | 640×480 | 0.005s | 77.0 | 1.47 |
| ✓ | 640×480 | 0.05s | 74.1 | 0.57 |

hand-object interactions and abrupt movements, which may not be fully captured by skeleton-based methods alone. The combination of these two modalities provides a comprehensive and robust understanding of human actions, leveraging the structural advantages of pose estimation and the dynamic sensitivity of neuromorphic vision.

## J    FUTURE WORK

**Real-World Multimodal Dataset Collection.** While our experiments rely on pseudo-event data synthesized from RGB via V2E, we acknowledge the limitations of such surrogate signals in modeling the stochastic, asynchronous nature of real Dynamic Vision Sensor (DVS) input. As a future direction, we plan to construct and release a real-world multimodal dataset comprising synchronized RGB streams, skeletal pose annotations, and event camera data. To this end, we will explore cross-sensor hardware synchronization, robust timestamp alignment, and markerless motion capture for skeleton labeling. Such a dataset would facilitate a more faithful evaluation of multimodal SNNs under real event data and encourage broader adoption of spiking paradigms in multimodal understanding.

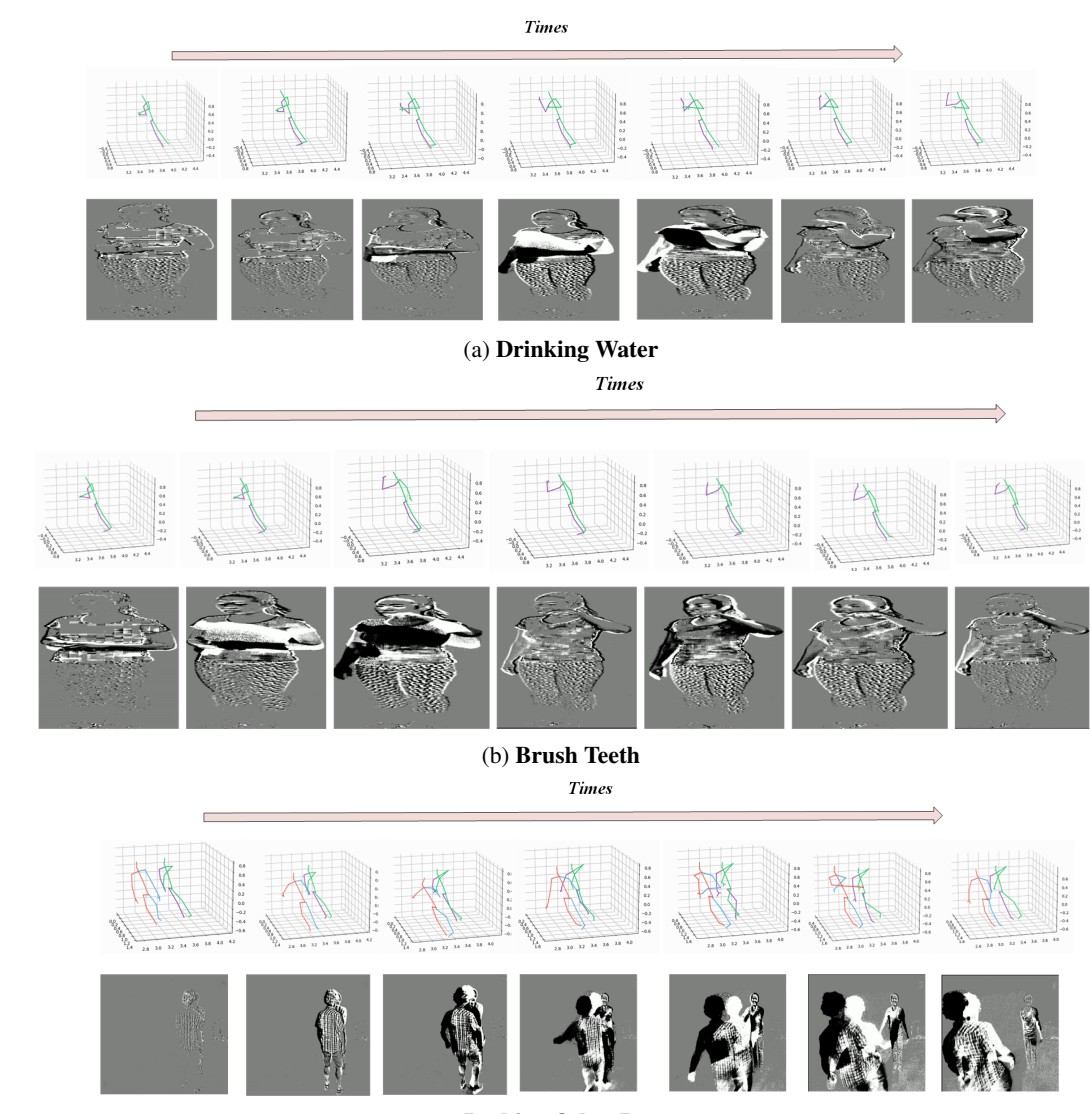

(a) **Drinking Water**

(b) **Brush Teeth**

(c) **Pushing Other Person**

Figure 11: Visualization of selected actions from NTU-RGB+D. (a) **Drinking water**. (b) **Brush Teeth**. (c) **Pushing other person**.

**Multimodal-Training, Unimodal-Inference (MT-UI).** In practical scenarios, it is often desirable to leverage rich multimodal supervision during training while retaining low-cost, single-modality inference at deployment. We intend to extend our model to support this asymmetrical paradigm—commonly referred to as Multimodal-Training, Unimodal-Inference (MT-UI), Learning Using Privileged Information (LUPI), or cross-modal distillation. Concretely, we will investigate: (i) modality dropout and consistency regularization to avoid co-adaptation during training; (ii) cross-modal teacher-student distillation where a spiking backbone learns to hallucinate or reconstruct representations of the missing modality; (iii) selective gating of modality-specific branches based on test-time availability and energy constraints. This line of work aims to enable robust and efficient action recognition even under partial modality conditions, while maintaining the benefits of joint training.

**Neuromorphic Deployment and Real-Time Evaluation.** Lastly, while we provide energy estimation through SOP-based proxy metrics, we plan to prototype our architecture on neuromorphic hardware platforms (e.g., Intel Loihi, DYNAP-SE, or FPGA-based accelerators) to validate real-

world latency, power, and throughput. This would offer a more practical perspective on deploying multimodal SNNs in mobile or wearable applications.

