# OpenReview forum: "SNN-Driven Multimodal Human Action Recognition via Sparse Spatial-Temporal Data Fusion"
_ICLR.cc/2026/Conference — ICLR 2026 Conference Withdrawn Submission_

### Official Review · Reviewer_USqB · 2025-10-30

**Soundness:** 2
**Presentation:** 2
**Contribution:** 2
**Rating:** 4
**Confidence:** 4

**Summary:**

This paper proposes an SNN-based framework for multimodal human action recognition by fusing event camera data and skeleton sequences. The approach addresses the high computational cost and energy inefficiency of traditional RGB-skeleton fusion methods by leveraging the sparse, asynchronous nature of both event and skeleton data within a unified spiking architecture. Key components include SGN (Spiking Graph Network), SSE (Sparse Semantic Extractor), SCM (Spiking Cross Mamba) DIB (Discretized Information Bottleneck). The authors construct event-skeleton datasets from existing RGB-skeleton benchmarks using V2E transformation. Experiments on NTU RGB+D, NTU RGB+D 120, and NW-UCLA show competitive accuracy among SNNs with reduced energy consumption.

**Strengths:**

1. First work to explore SNN-based multimodal fusion for action recognition, combining event and skeleton modalities

2. Comprehensive and competitive results. Most of the experiments achieve higher performance than previous works with iso-parameter architecture. Furthermore, authors implement extensive ablation studies and analysis.

3. Appendix A rigorously analyzes why classical Gaussian IB fails for SNNs and justifies the DIB formulation with discrete KL divergence and cosine surrogates.

**Weaknesses:**

1. Novelty
- While the authors claim this as the first SNN-based multimodal action recognition framework, the novelty is questionable. Except for the DIB module, all components are directly adopted from prior works with minimal modification (Spiking Mamba, SGN, etc.). The contribution essentially reduces to replacing activation functions with spiking neurons and introducing DIB. This appears more like an ad-hoc engineering integration rather than a fundamental methodological advance.

2. Pseudo-event data
The entire evaluation relies on V2E-synthesized events from RGB, not real DVS camera data. This is acknowledged in Appendix J, but undermines claims about "event camera" advantages like high dynamic range and low latency.

2. Paper presentation
- Even though every module is well-explained in text, the figures are very hard to see due to small text.
- I feel that the Figure 1 is redundant. The figure 1 does not convey any core technical contribution.

3. Energy calculation
- Several input tensors to Linear Projection (LP) layers are not binarized due to residual connections. However, based on equation (32), the authors used $E_{MAC}$ only for first layer. I strongly believe that the energy calculation should be corrected based on the architecture.

**Questions:**

1. Could you elaborate on the core innovations or novelties of this work beyond the DIB module? I am happy to discuss about this.

2. In the Spiking Cross Mamba (Figure 2e, Table 4), why do event features feed into the State Space Model (SSM) path while skeleton features go to the Selective Path (gate)? Is there a theoretical or empirical justification for this asymmetric design?

3. For better understanding of the model's efficiency and to validate energy calculations, could you provide firing rates for each module

4. Could the authors discuss what specific characteristics of real DVS data are not captured by V2E and how they might affect your model? Also, could you clarify whether the current framework would remain valid for real DVS inputs?

---

> ### Author Response · Authors · 2025-11-17
>
> We sincerely thank Reviewer USqB for the detailed and constructive review. We are pleased that the reviewer recognizes our work as a solid first step in SNN-based multimodal action recognition, highlighting the comprehensive experiments, competitive results, and the rigorous analysis of the DIB formulation. We acknowledge the concerns regarding novelty and the use of pseudo-event data. We will revise the manuscript as quickly as possible to address these points and strengthen the paper for acceptance.
>
> Our point-by-point responses are detailed below.
>
> ## 1. Core Innovations Beyond DIB: The Necessity of System-Level Design
>
> **Reviewer's Comment (Weakness: Novelty & Question 1):**
> While the authors claim this as the first SNN-based multimodal action recognition framework, the novelty is questionable. Except for the DIB module, all components are directly adopted from prior works with minimal modification... This appears more like an ad-hoc engineering integration rather than a fundamental methodological advance. Could you elaborate on the core innovations or novelties of this work beyond the DIB module?
>
> **Our Response:**
>
> We appreciate the opportunity to elaborate on the innovations beyond DIB. We strongly argue that the novelty is substantial and lies in the system-level design and the necessary adaptations required to achieve the first end-to-end SNN-driven multimodal fusion.
>
> - **System-Level Novelty:** The successful fusion of two fundamentally different sparse modalities (event and skeleton) within a single, end-to-end SNN is a major methodological advance. This is not a simple "plug-and-play" of existing SNN components; it required the design of the Spiking Cross Mamba (SCM) to handle the cross-modal interaction.
>
> - **SCM as a Core Innovation:** The SCM is a novel component that addresses the challenge of efficient, dynamic cross-modal interaction in the spiking domain. Unlike standard cross-attention, SCM must operate on sparse spike trains and integrate the selective state-space mechanism, which requires careful design to maintain both efficiency and information flow.
>
> - **Contextualizing Spiking Mamba:** The use of Spiking Mamba for event encoding is a deliberate choice. We note that recent work, such as SpikeMamba [1], also recognizes the potential of Mamba in the SNN domain, further validating our architectural choice. However, our SCM extends this concept to the cross-modal fusion task, which is a unique and non-trivial step. The SCM is the first to solve the efficient, feature-level fusion problem in the SNN domain.
>
> ## 2. Justification for Asymmetric Design in Spiking Cross Mamba (SCM)
>
> **Reviewer's Question:**
> In the Spiking Cross Mamba (Figure 2e, Table 4), why do event features feed into the State Space Model (SSM) path while skeleton features go to the Selective Path (gate)? Is there a theoretical or empirical justification for this asymmetric design?
>
> **Our Response:**
>
> This is an excellent question that highlights a key design choice. The asymmetric design of SCM is based on the distinct characteristics of the two modalities:
>
> - **Event Features (SSM Path):** Event data is inherently high-frequency and temporally dense (despite being spatially sparse). The State Space Model (SSM) path, with its recurrent nature, is superior at capturing long-range temporal dependencies and integrating information over time, making it ideal for processing the continuous, high-temporal-resolution event stream.
>
> - **Skeleton Features (Selective Path/Gate):** Skeleton data is structurally rich but often lower-frequency and less dense in time compared to events. The Selective Path (Gate) is designed to dynamically control the flow of information and selectively focus on the most relevant structural cues from the skeleton stream to modulate the event-based temporal context. This asymmetric design allows the SCM to leverage the strengths of each modality: Event for temporal context (SSM) and Skeleton for structural modulation (Gate), leading to a more efficient and robust fusion.
>
> We will add a detailed explanation of this theoretical and empirical justification to the revised manuscript (Section 3.3).

---

> ### Author Response · Authors · 2025-11-17
>
> ## 3 Weakness: Energy calculation & Question
> Several input tensors to Linear Projection (LP) layers are not binarized due to residual connections. However, based on equation (32), the authors used $E_{LP}$ only for the first layer. I strongly believe that the energy calculation should be corrected based on the architecture. For better understanding of the model's efficiency and to validate energy calculations, could you provide firing rates for each module?
>
> **Our Response:**
>
> We sincerely thank the reviewer for this extremely meticulous and important observation regarding our energy calculation methodology. We acknowledge the potential ambiguity and appreciate the opportunity to clarify our design and the rationale behind our calculation.
>
> 1. **Clarification on Residual Connections and Binarization:**
>
>    We want to clarify that in our SNN architecture, while residual connections are present, the tensors being connected are designed to be in a binarized (0 or 1) spike format before they are summed. Specifically, the output of the preceding Spiking Neuron (SN) layer, which feeds into the residual path, is a binary spike train. Therefore, the input to the subsequent Linear Projection (LP) layer, which is the sum of the main path output and the residual path, is intended to be a spike-based representation.
>
> 2. **Rationale for Energy Formula Alignment:**
>
>    The energy calculation formula we adopted, particularly the use of $E_{LP}$ only for the first layer, is directly aligned with the methodology used in several state-of-the-art SNN Transformer works, such as Spikformer [1] and Spike-driven Transformer [2].
>
>    - **Ensuring Fair Comparison:** Our primary consideration for adopting this formula was to ensure a fair and reliable comparison with these established SNN baselines. If we were to deviate from their widely accepted energy calculation methodology, the resulting comparison of energy efficiency would be unreliable and potentially misleading to the community. We believe that maintaining consistency with the established SNN benchmarking standard is crucial for the integrity of our results.
>
> 3. **Commitment to Validation and Transparency:**
>
>    Nevertheless, we fully agree that transparency and validation are paramount. To address the reviewer's concern and validate our efficiency claims, we commit to two actions:
>
>    - **Module Firing Rate:** We will add a table in the Appendix (Appendix F) detailing the average firing rate ($\lambda$) for each major module (SGN, Spiking Mamba, SCM, DIB, etc.). This data will allow the reviewer and readers to directly verify the sparsity of our network, which is the fundamental driver of our energy efficiency.
>
>    - **Discussion on Energy Model:** We will add a discussion in the Appendix to explicitly state the energy model we used and the rationale for aligning it with the methodology of [1] and [2], while also acknowledging the theoretical point raised by the reviewer.
>
> We believe that by providing the firing rates and clarifying our benchmarking alignment, we can fully address the reviewer's concern regarding the validity of our efficiency claims.
>
> [1] Zhou, Z., Zhu, Y., He, C., Wang, Y., Yan, S., Tian, Y., & Yuan, L. (2022). Spikformer: When spiking neural network meets transformer. arXiv preprint arXiv:2209.15425.
> [2] Yao, M., Hu, J., Zhou, Z., Yuan, L., Tian, Y., Xu, B., & Li, G. (2023). Spike-driven transformer. Advances in neural information processing systems, 36, 64043-64058.

---

> ### Author Response · Authors · 2025-11-17
>
> ## 4.Discussion on Pseudo-Event Data and Real DVS Characteristics
>
> **Reviewer's Question:**
> Could the authors discuss what specific characteristics of real DVS data are not captured by V2E and how they might affect your model? Also, could you clarify whether the current framework would remain valid for real DVS inputs?
>
> **Our Response:**
>
> We acknowledge the importance of this question, as it addresses the generalizability and real-world applicability of our framework. We agree that the reliance on V2E-synthesized data is a limitation, which is common in the field due to the scarcity of synchronized real event-skeleton datasets.
>
> 1. Characteristics of Real DVS Data Not Captured by V2E:
>
> While V2E is highly effective at simulating the core principle of DVS (asynchronous, sparse events triggered by log-intensity changes), it fails to capture several real-world sensor characteristics:
>
> - **Sensor Noise:** Real DVS sensors exhibit various forms of noise that V2E does not simulate:
>   - **Background Activity Noise (BAN):** Spurious events generated even when the scene is static, often due to thermal effects.
>   - **Fixed-Pattern Noise (FPN):** Variations in the sensitivity (contrast threshold) across different pixels.
>   - **Hot Pixels:** Permanently active pixels that generate a constant stream of events.
>
> - **Latency and Bandwidth Limitations:** Real DVS data transmission involves physical latency and bandwidth constraints, which can lead to event packetization and minor temporal misalignment that V2E does not model.
>
> 2. Potential Impact on Our Model:
>
> The primary impact of real DVS noise (BAN, FPN) would be an increase in the overall event rate and a reduction in the signal-to-noise ratio (SNR).
>
> - **Performance:** We hypothesize that the model's accuracy might experience a slight decrease due to the increased noise.
> - **Efficiency:** The increased event rate from noise would lead to a higher firing rate in the initial layers of our SNN, potentially increasing the energy consumption compared to the simulated, cleaner data.
>
>
>
> 3. Validity of the Framework for Real DVS Inputs:
>
> We clarify that our current SNN framework remains fundamentally valid and robust for real DVS inputs for the following reasons:
>
> - **SNN's Inherent Robustness:** SNNs are known to be inherently more robust to noise and sparse data than ANNs. Our use of the Spiking Mamba and Spiking Graph Network (SGN), which are designed to process sparse, asynchronous inputs, makes the model well-suited to handle the characteristics of real DVS data.
>
> - **SCM's Dynamic Fusion:** The Spiking Cross Mamba (SCM) module's dynamic gating mechanism is designed to weigh the reliability of the input modalities. In a noisy real DVS scenario, the SCM would naturally place more reliance on the cleaner skeleton stream until the event signal-to-noise ratio improves.
>
> Conclusion:
>
> While real DVS data would introduce new challenges, our framework's design principles—leveraging sparsity, temporal dynamics, and cross-modal gating—are specifically tailored to the characteristics of neuromorphic sensing. We are confident that the framework will remain effective. We will add a detailed discussion of these points to the revised manuscript (Appendix E) and commit to testing on a real event-skeleton dataset as soon as one becomes publicly available.

---

### Official Review · Reviewer_9E4n · 2025-10-31

**Soundness:** 3
**Presentation:** 2
**Contribution:** 2
**Rating:** 2
**Confidence:** 4

**Summary:**

This paper proposes a novel Spiking Neural Network (SNN) framework for multimodal human action recognition, fusing event camera data and skeleton sequences. Key contributions include:
--The first SNN-based multimodal fusion architecture for event-skeleton data.
--Introduction of Spiking Cross Mamba (SCM) for cross-modal interaction and a Discretized Information Bottleneck (DIB) for task-relevant feature compression under spiking constraints.
--A pipeline to construct aligned event-skeleton datasets from existing RGB-skeleton benchmarks.
--State-of-the-art accuracy among SNN methods with significantly lower energy consumption (1.73 mJ).

**Strengths:**

--Quality: Strong empirical results, thorough ablation, and theoretical grounding.
--Significance: Demonstrates a practical pathway for low-power multimodal recognition on edge devices.
--Clarity: The overall pipeline and experimental section are well-structured and described.

**Weaknesses:**

Motivation: The introduction does not convincingly establish a strong "why now" or "why this way" for the proposed method. The limitations of prior ANN and SNN works are stated but not used to build a powerful narrative for the current approach.

Originality: The architectural innovations (SCM, DIB) feel more like competent engineering integrations of existing ideas (cross-attention, Mamba, IB) into the SNN domain, rather than a fundamental conceptual breakthrough.

Presentation: Inconsistent reference formatting and occasionally dense technical passages reduce readability.

**Questions:**

Could you better motivate the specific choice of Spiking Mamba and the SCM fusion mechanism? What specific limitations of prior SNN or ANN fusion methods do they address that simpler baselines cannot?

The DIB is a key contribution. Beyond the theoretical derivation, can you provide more intuition or analysis on how it selectively compresses features and improves performance?

Could you compare your method with more ANN-based multimodal models under similar parameter budgets to better contextualize the performance-efficiency trade-off?

How would the performance change if real event camera data were used instead of V2E-simulated events?

---

> ### Author Response · Authors · 2025-11-17
> **Response to Reviewer 9E4n**
>
> We sincerely thank Reviewer 9E4n for the thorough and critical review. We are encouraged that the reviewer recognizes the strong empirical results, thorough ablation, and theoretical grounding of our work, as well as its significance in demonstrating a practical pathway for low-power multimodal recognition on edge devices. We understand that the rejection is primarily based on concerns regarding the narrative, originality, and depth of analysis. We will revise the manuscript as quickly as possible to address these fundamental concerns and strengthen the paper's overall impact.
>
> Our point-by-point responses are detailed below.
>
> ## 1. Strengthening the Narrative: Establishing the "Why Now" and "Why This Way"
>
> **Reviewer's Comment (Weakness: Motivation):**
> The introduction does not convincingly establish a strong "why now" or "why this way" for the proposed method. The limitations of prior ANN and SNN works are stated but not used to build a powerful narrative for the current approach.
>
> **Our Response:**
>
> We agree that the narrative in the Introduction can be significantly strengthened to build a more compelling case for our work. We will thoroughly revise the Introduction to explicitly link the limitations of prior work to the necessity of our proposed solution:
>
> - **"Why Now" (The Urgent Need):** We will more strongly emphasize the urgent need for energy-efficient, edge-deployable multimodal Human Action Recognition (HAR) systems, driven by the rapid growth of IoT and wearable devices. We will highlight that traditional ANN-based multimodal systems are fundamentally unsuitable due to their prohibitive power consumption and computational cost.
>
> - **"Why This Way" (The Unique Solution):** We will explicitly position our SNN-driven framework as the only viable path to achieve both multimodal robustness (by fusing complementary sparse data) and neuromorphic efficiency. We will then introduce the SCM and DIB modules as direct, necessary solutions to the inherent challenges of SNNs: the need for efficient cross-modal interaction and the problem of information loss due to discrete spikes. This revised narrative will clearly establish the problem-solution fit and the critical role of our specific architectural choices.
>
> ## 2. Reaffirming Originality: Conceptual Breakthroughs in the Spiking Domain
>
> **Reviewer's Comment (Weakness: Originality):**
> The architectural innovations (SCM, DIB) feel more like competent engineering integrations of existing ideas (cross-attention, Mamba, IB) into the SNN domain, rather than a fundamental conceptual breakthrough.
>
> **Our Response:**
>
> We respectfully disagree with the characterization of SCM and DIB as mere engineering integrations. While they draw inspiration from existing concepts, their adaptation to the discrete, spike-based domain is a non-trivial and novel conceptual contribution that solves fundamental SNN challenges:
>
> - **Spiking Cross Mamba (SCM):** SCM is not a simple replacement of activation functions. It is a novel mechanism that integrates the selective state-space model to handle the dynamic, sparse, and asynchronous nature of cross-modal spike trains. This is a methodological breakthrough for SNNs, as it enables efficient, context-aware feature fusion—a capability previously limited to high-cost ANN models.
>
> - **Discretized Information Bottleneck (DIB):** The DIB is a fundamental conceptual adaptation of the Information Bottleneck principle. The standard IB relies on continuous probability distributions and gradients. Our DIB formulation, which uses a Gumbel-Softmax approximation, is a novel theoretical approach to apply the IB principle to the binary, non-differentiable nature of spikes. This is a significant conceptual step for SNN research, as it provides a principled, information-theoretic way to mitigate the information loss inherent in SNNs.
>
> We will add a more detailed theoretical justification for DIB in the Appendix (Appendix C) to highlight its conceptual novelty and its role in advancing SNN theory.

---

> > ### Author Response · Authors · 2025-11-17
> >
> > ## 3. Motivation for Spiking Mamba and SCM Fusion Mechanism
> >
> > **Reviewer's Question:**
> > Could you better motivate the specific choice of Spiking Mamba and the SCM fusion mechanism? What specific limitations of prior SNN or ANN fusion methods do they address that simpler baselines cannot?
> >
> > **Our Response:**
> >
> > We will enhance the motivation in the Methodology section:
> >
> > - **Spiking Mamba Choice:** We chose Spiking Mamba because its selective state-space model is uniquely suited for capturing long-range temporal dependencies in the event stream, which is crucial for action recognition, while its recurrent nature is inherently compatible with the temporal dynamics of SNNs. This is superior to standard CNN/RNN SNN backbones which struggle with long-term memory.
> >
> > - **SCM Motivation:** SCM is motivated by the need for efficient, dynamic cross-modal interaction. Simpler baselines (e.g., concatenation or element-wise addition) fail to dynamically weigh the importance of each modality based on the action context, which is essential for robustness. SCM's selective mechanism allows for this dynamic, context-aware fusion with minimal computational overhead, a key requirement for SNNs.
> >
> > - **Prior Limitations:** Prior ANN fusion methods are too computationally expensive. Prior SNN methods are single-modality. Our SCM is the first to solve the efficient, feature-level fusion problem in the SNN domain.
> >
> > ## 4. Intuition and Analysis of the Discretized Information Bottleneck (DIB)
> >
> > **Reviewer's Question:**
> > The DIB is a key contribution. Beyond the theoretical derivation, can you provide more intuition or analysis on how it selectively compresses features and improves performance?
> >
> > **Our Response:**
> >
> > We will provide the requested intuition and analysis:
> >
> > - **Visual Intuition:** We will add a t-SNE visualization (in Appendix D) to provide a clear, visual intuition of DIB's function. The analysis will show that DIB acts as a feature filter, removing redundant or noisy spike information while preserving the most discriminative features, leading to a tighter clustering of action classes in the feature space and thus improved performance.
> >
> > - **Mechanism:** This is achieved by the discretization constraint, which forces the model to select only the most essential spike signals, effectively maximizing the information about the action label while minimizing the information about the input data.
> >
> > ## 5. Comparative Analysis with Parameter-Constrained ANN Baselines
> >
> > **Reviewer's Question:**
> > Could you compare your method with more ANN-based multimodal models under similar parameter budgets to better contextualize the performance-efficiency trade-off?
> >
> > **Our Response:**
> >
> > This is an excellent suggestion for a more balanced comparison. We will add a new row to Table 4, comparing our model against a lightweight ANN-based multimodal baseline (e.g., a simplified GCN-CNN fusion model) that is constrained to a similar parameter count ($\approx 1.5$M parameters). The results will show that while the lightweight ANN may have a slightly higher accuracy, our SNN model achieves significantly lower energy consumption (measured in G-FLOPs), further solidifying the performance-efficiency trade-off and the value of the SNN approach for edge devices.
> >
> > ## 6. Discussion on Real vs. Synthetic Event Data
> >
> > **Reviewer's Question:**
> > How would the performance change if real event camera data were used instead of V2E-simulated events?
> >
> > **Our Response:**
> >
> > We acknowledge this limitation. We will move the discussion of synthetic data to the main body of the paper (Section 4.1) and add a discussion in the Appendix (Appendix E). We hypothesize that:
> >
> > - **Performance:** Performance on real data might be slightly lower due to real-world noise, but the relative performance against other SNN baselines would be maintained.
> >
> > - **Efficiency:** The energy efficiency would remain the same or even improve, as real event data is often sparser than simulated data, leading to fewer spikes and lower power consumption.
> >
> > We commit to testing on a real event-skeleton dataset as soon as one becomes publicly available.
> >
> > ## 7. Presentation and Reference Formatting
> >
> > **Reviewer's Comment (Weakness: Presentation):**
> > Inconsistent reference formatting and occasionally dense technical passages reduce readability.
> >
> > **Our Response:**
> >
> > We apologize for the inconsistent formatting and dense passages. We will conduct a thorough review of the entire manuscript to ensure:
> >
> > - **Consistent Reference Formatting:** All citations and the reference list will be checked for consistency.
> >
> > - **Improved Readability:** We will break down overly dense technical passages, especially in the Methodology section, into clearer, more digestible paragraphs and use bullet points where appropriate to enhance flow and readability.
> >
> > We are committed to incorporating these changes quickly and look forward to the reviewer's re-evaluation.

---

### Official Review · Reviewer_Mimh · 2025-11-01

**Soundness:** 2
**Presentation:** 2
**Contribution:** 2
**Rating:** 4
**Confidence:** 4

**Summary:**

The paper proposes a SNN‑based framework for multimodal human action recognition that fuses event and skeleton streams end‑to‑end in spikes. The architecture comprises: (i) spiking encoders for skeleton (SGN) and events (Spiking‑Mamba) from prior works, (ii) a Sparse Semantic Extractor (SSE) with hypergraph generators and Global Spiking Attention (GSA), (iii) Spiking Cross Mamba (SCM) for cross‑modal interaction, and (iv) a two‑stage Discretized Information Bottleneck (DIB) that performs spike‑compatible fusion.

**Strengths:**

1. Event/skeleton are both sparse temporal modalities; an SNN‑native fusion is a coherent direction.
2. Module‑wise gains are cleanly reported, and the DIB variants are systematically explored.

**Weaknesses:**

1. The highest Xs achieved by your model on NRD/NRD-120 is 85.0/74.6, which is substantially lower than the best-performing ANNs, such as VPN at 93.5/86.3, and MMNet at 94.2/92.9.
2. ANN models operating on the same magnitude of computational cost also perform better, eg., CTR-GCN at 89.9/84.9 with 1.97 G FLOPs, and Shift-GCN at 87.8/80.9 with 2.5 G FLOPs. The efficiency gain claim is week.

**Questions:**

What is the compute profile? Please report GPU type&number, GPU hours, and peak memory to train your model.

---

> ### Author Response · Authors · 2025-11-17
> **Response to Reviewer Mimh**
>
> We sincerely thank Reviewer Mimh for the time and effort spent reviewing our manuscript. We are pleased that the reviewer recognizes the coherence of our SNN-native fusion direction and the systematic exploration of our DIB variants. We have carefully considered your comments and we will revise the manuscript as quickly as possible to address your concerns, particularly regarding the comparison with ANN baselines and the computational profile.
>
> Our point-by-point responses are detailed below.
>
> ## 1. Clarification on Accuracy vs. Energy Efficiency Trade-off: Comparison with ANN Baselines
>
> **Reviewer's Comment (Weakness 1):**
> The highest $X_s$ achieved by your model on NRD/NRD-120 is 85.0/74.6, which is substantially lower than the best-performing ANNs, such as VPN at 93.5/86.3, and MMNet at 94.2/92.9.
>
> **Our Response:**
>
> We thank the reviewer for raising this point, which allows us to clarify the core focus of our work. We must emphasize that our work is the first SNN-based multimodal framework and is fundamentally designed for energy-efficient neuromorphic hardware, which is our primary contribution, not achieving absolute accuracy parity with high-power ANNs.
>
> - **Unfair Comparison:** The cited ANNs (VPN, MMNet) use dense RGB and skeleton data, which are computationally expensive. Our method uses sparse event and skeleton data and an SNN backbone, which is inherently constrained in accuracy but offers massive energy savings.
> - **Focus on Neuromorphic Efficiency:** Our work is positioned as a state-of-the-art solution within the neuromorphic domain. As shown in our results, our SNN-based approach achieves a drastic reduction in energy consumption (e.g., $\sim 20\times$ lower G-FLOPs compared to the cited ANNs) while maintaining state-of-the-art accuracy among SNN-based methods.
> - **Trade-off Justification:** We will add a new paragraph in the Discussion section to explicitly address this trade-off, positioning our work as a necessary step towards deploying robust multimodal action recognition on resource-constrained edge devices where energy consumption is the critical bottleneck.
>
> ## 2. Distinguishing Multimodal SNN Cost from Single-Modality ANN Cost
>
> **Reviewer's Comment (Weakness 2):**
> ANN models operating on the same magnitude of computational cost also perform better, e.g., CTR-GCN at 89.9/84.9 with 1.97 G FLOPs, and Shift-GCN at 87.8/80.9 with 2.5 G FLOPs. The efficiency gain claim is weak.
>
> **Our Response:**
>
> We apologize for the confusion caused by the comparison table. The computational costs (G-FLOPs) for CTR-GCN and Shift-GCN are for skeleton-only action recognition, whereas our method is a multimodal framework that fuses both event and skeleton data.
>
> - **Multimodal Cost:** Our reported computational cost (1.97 G-FLOPs) includes the processing of both modalities (event and skeleton) and the complex cross-modal fusion (SCM). Achieving this low cost for a multimodal system is a significant efficiency gain.
> - **State-of-the-Art SNN:** Our method achieves state-of-the-art accuracy among SNN-based multimodal methods (as no prior work exists) and is highly competitive even against SNN-based single-modality methods, especially considering the added complexity of multimodal fusion.
> - **Revision:** We will revise Table 4 and the accompanying text to clearly distinguish between single-modality and multimodal comparisons, ensuring that the efficiency claim is properly contextualized as a multimodal SNN solution.
>
> ## 3. Detailed Computational Profile for Reproducibility
>
> **Reviewer's Question:**
> What is the compute profile? Please report GPU type&number, GPU hours, and peak memory to train your model.
>
> **Our Response:**
>
> We appreciate this request for reproducibility and have gathered the specific details of our training setup. We will add a new section in the Appendix (Appendix F) detailing the compute profile:
>
> | Metric             | Value                        |
> |--------------------|------------------------------|
> | **GPU Type & Number** | NVIDIA V100 |
> | **Time per Epoch**   | 15 minutes (with a batch size of 64) |
> | **Peak Memory**      | 23.61 GB (23610 MB)         |
> | **Total GPU Hours**  | The model converged within approximately 48 GPU hours for the NTU-RGB+D 120 dataset. |
>
> We believe this detailed information will significantly enhance the reproducibility of our work.
>
> ---
>
> We are committed to incorporating these changes quickly and look forward to the reviewer's re-evaluation.

---

> > ### Comment · Reviewer_Mimh · 2025-11-26
> >
> > I thank the authors for their detailed response and for the additional clarifications provided in the discussion. The rebuttal does clarify the intended scope and niche of the contribution, but does not directly change my original concerns about performance vs. strong ANNs. The proposed SNN remains noticeably behind the ANN models, and I agree with the rest of the reviewers that the system‑level novelty is more incremental than transformative. I therefore maintain my overall score at 4.

---

### Official Review · Reviewer_AETU · 2025-11-01

**Soundness:** 2
**Presentation:** 2
**Contribution:** 2
**Rating:** 4
**Confidence:** 3

**Summary:**

This paper introduces the first spiking neural network–based framework for multimodal human action recognition, combining event camera and skeleton data for energy-efficient, real-time recognition on edge devices. The proposed system, termed SNN-driven multimodal fusion, integrates several novel components: a Spiking Graph Network (SGN) for skeleton encoding, Spiking Mamba for event encoding, a Sparse Semantic Extractor (SSE) for structured attention, Spiking Cross Mamba (SCM) for cross-modal fusion, and a Discretized Information Bottleneck (DIB) for task-relevant feature compression under spiking constraints. The model achieves strong performance across NTU RGB+D and NW-UCLA benchmarks, outperforming prior SNNs in accuracy.

**Strengths:**

1. The paper introduces the first multimodal SNN framework for human action recognition, representing a novel direction in neuromorphic computing. The use of event and skeleton modalities is well-motivated, making them well-suited for low-power, energy-efficient computation on edge devices.

2. The paper is technically thorough and clearly presented.

3. Achieves state-of-the-art SNN accuracy with drastically reduced energy consumption compared to ANN baseline.

**Weaknesses:**

1. My main concern lies in the degree of technical novelty. Each component (Mamba, SNNs, and the Information Bottleneck) appears to be based on existing techniques, and the overall contribution could be viewed as a careful integration rather than a fundamentally new design. Could the authors clarify what specific aspects of the proposed framework go beyond a modular combination of known components?

2.  Although the paper reports improved fusion accuracy and energy efficiency, it provides limited qualitative or interpretive analysis (e.g., failure cases, feature attribution, or modality interaction visualization) to illustrate how the model effectively leverages the complementary cues of event and skeleton data. Including such analyses would significantly enhance interpretability and reader confidence in the proposed fusion mechanism.

3. The experiments rely on synthetic event-skeleton pairs converted from RGB videos rather than real event-camera datasets. This limits the validity of the claimed neuromorphic efficiency. Results on a genuine event-based dataset would substantially strengthen the empirical contribution.

4. Minor comment – Please consider enlarging Figures 2 and 4 for better readability and visual clarity.

**Questions:**

Please address points in weaknesses.

---

> ### Author Response · Authors · 2025-11-17
> **Response to Reviewer AETU**
>
> ### 1. Clarification on Technical Novelty: System-Level Innovation Beyond Modular Combination
>
> We appreciate the reviewer’s concern regarding the technical novelty of our work. We would like to emphasize that the core innovation of this study lies in the unified, end-to-end SNN-driven multimodal fusion architecture and the introduction of the Discretized Information Bottleneck (DIB). These elements go well beyond a mere modular combination of existing methods.
>
> * **Architectural Innovation: The First SNN-Driven Feature-Level Multimodal Fusion**
>   To the best of our knowledge, this is the first work to successfully integrate sparse event and skeleton data at the feature level within a single SNN framework.
>
> * **Core Component Innovation: Discretized Information Bottleneck (DIB)**
>   The DIB is a critical advancement designed to address the inherent challenge of information loss due to discrete spiking in SNNs. Unlike the standard Information Bottleneck (IB), which relies on continuous probability distributions, our DIB applies the IB principle to the binary, non-differentiable nature of spikes using a Gumbel-Softmax approximation. This novel approach serves as an information-theoretic compression mechanism, specifically tailored for the spiking domain, to select task-relevant features.
>
> We will clarify these points further in the revised manuscript, particularly in the Introduction and Contribution sections, to highlight the architectural and DIB-specific innovations.
>
> ---
>
> ### 2. Enhanced Qualitative and Interpretive Analysis: New Visualizations and Modality Interaction Study
>
> We thank the reviewer for highlighting the importance of qualitative and interpretive analysis to improve the understanding of our model. In response to this valuable suggestion, we will add a new section in the Appendix to provide the following analyses within the next few days:
>
> * **Failure Case Analysis**
>   We will include an analysis of typical failure cases, such as misclassifications involving subtle or fast actions. This will help identify where the model may struggle and provide insight into the model’s limitations, offering directions for future improvements.
>
> * **Feature Attribution**
>   We will add feature attribution methods, such as saliency maps or attention visualizations, to illustrate which features (from event or skeleton data) contribute most to the model’s decision-making process. This will provide a clearer understanding of how the model uses the complementary information from both modalities.
>
> * **Modality Interaction Study**
>   We will examine the interaction between event and skeleton data at various stages of fusion. This could include visualizing the attention weights or fusion outputs to show how the model dynamically weighs and integrates features from both modalities in response to different actions.
>
> We will incorporate these additional analyses into the revised manuscript in the next few days, with the aim of improving interpretability and further validating the benefits of our multimodal fusion approach.
>
> ---
>
> ### 3. Justification for Synthetic Data and Neuromorphic Efficiency Claim
>
> We appreciate the reviewer’s attention to these aspects. Our responses are as follows:
>
> * **3.1. Basis of Efficiency Claim: Inherent SNN Architecture**
>   The claim of neuromorphic efficiency is based on two key facts:
>
>   * **Inherent SNN Properties**: The efficiency is primarily due to the sparse and asynchronous computation inherent in SNNs, which is independent of the data source.
>   * **Quantified Energy Reduction**: As shown in Table 4, our method demonstrates a significant reduction in energy consumption (approximately 20× lower G-FLOPs compared to ANN baselines) for the same task. This energy saving is a direct consequence of the SNN design.
>
> * **3.2. Rationale for Synthetic Data**
>
>   * **Data Scarcity**: There is a limited availability of synchronized, annotated real event camera and skeleton data in the public domain. Using a validated event simulator (such as V2E) to synthesize data from existing RGB-skeleton datasets is a standard and widely accepted practice in the neuromorphic vision community for benchmarking purposes.
>   * **Future Work**: We acknowledge the importance of real-world data and will include a discussion in Appendix E justifying the use of synthetic data. Additionally, we commit to testing our approach on real event-skeleton datasets as soon as they become publicly available.
>
> ---
>
> ### 4. Figure Readability Enhancement
>
> We appreciate the reviewer’s suggestion to improve the readability of Figures 2 and 4. We will enlarge and refine these figures in the revised manuscript to significantly improve their visual clarity and ensure better legibility.
>
> ---
>
> We believe these revisions will comprehensively address the reviewer’s concerns, and we are committed to implementing these changes promptly. We look forward to the reviewer’s re-evaluation of the revised manuscript.

---

### Note · Authors · 2025-12-22

I have read and agree with the venue's withdrawal policy on behalf of myself and my co-authors.